# PIP4K2B is mechanoresponsive and controls heterochromatin-driven nuclear softening through UHRF1

Alessandro Poli[1] ✉, Fabrizio A. Pennacchio[1], Andrea Ghisleni[1], Mariagrazia di Gennaro[1], Margaux Lecacheur [1], Paulina Nastaly [2], Michele Crestani[1], Francesca M. Pramotton [3,4], Fabio Iannelli [1], Galina Beznusenko [1], Alexander A. Mironov[1], Valeria Panzetta[5,6,7], Sabato Fusco[8], Bhavwanti Sheth[9], Dimos Poulikakos [4], Aldo Ferrari[4], Nils Gauthier[1], Paolo A. Netti[5,6,7], Nullin Divecha [9] & Paolo Maiuri [1,10] ✉

Phosphatidylinositol-5-phosphate−4-kinases (PIP4Ks) are stress-regulated phosphoinositide kinases able to phosphorylate PtdIns5P to PtdIns(4,5)P2. In cancer patients their expression is typically associated with bad prognosis. Among the three PIP4K isoforms expressed in mammalian cells, PIP4K2B is the one with more prominent nuclear localisation. Here, we unveil the role of PIP4K2B as a mechanoresponsive enzyme. PIP4K2B protein level strongly decreases in cells growing on soft substrates. Its direct silencing or pharmacological inhibition, mimicking cell response to softness, triggers a concomitant reduction of the epigenetic regulator UHRF1 and induces changes in nuclear polarity, nuclear envelope tension and chromatin compaction. This substantial rewiring of the nucleus mechanical state drives YAP cytoplasmic retention and impairment of its activity as transcriptional regulator, finally leading to defects in cell spreading and motility. Since YAP signalling is essential for initiation and growth of human malignancies, our data suggest that potential therapeutic approaches targeting PIP4K2B could be beneficial in the control of the altered mechanical properties of cancer cells.

Mechanosensing impacts on a variety of processes, regulating genome integrity, gene expression and cell migration[1,2]. Deregulated mechanotransduction is often associated with the insurgence of different pathologies, including cancer[2–5]. Recent studies highlighted the mechanosensing role of the nucleus. It acts, indeed, as an intracellular ruler of mechanically induced cellular shape variations. Mechanical forces acting on the cell and then on the nucleus drive changes in nuclear envelope (NE) composition and chromatin organisation, then determining downstream cellular responses[2–5].

Polyphosphoinositides (PPIns/PI) are lipid second messengers involved in many cellular functions, including proliferation, adhesion, cytoskeletal organisation and transcription[6]. Nuclear PIs, and more

[1]IFOM ETS - The AIRC Institute of Molecular Oncology, Milan, Italy. [2]Laboratory of Translational Oncology, Intercollegiate Faculty of Biotechnology, University of Gdańsk and Medical University of Gdańsk, Gdansk, Poland. [3]EMPA-Materials Science and Technology, Dubenforf, Switzerland. [4]Institute for Mechanical Systems, ETH, Zurich, Switzerland. [5]Department of Chemical, Materials and Production Engineering, University of Naples Federico II, Naples, Italy. [6]Centro di Ricerca Interdipartimentale sui Biomateriali CRIB, University of Naples Federico II, Naples, Italy. [7]Istituto Italiano di Tecnologia, IIT@CRIB, Naples, Italy. [8]Department of Medicine and Health Sciences "V. Tiberio", University of Molise, Campobasso, Italy. [9]Inositide Laboratory, School of Biological Sciences, Faculty of Environmental and Life Sciences, University of Southampton, Southampton, United Kingdom. [10]Department of Molecular Medicine and Medical Biotechnology, University of Naples Federico II, Naples, Italy. ✉e-mail: allen2315@gmail.com; paolo.maiuri@unina.it

generally proteins containing PI-interacting domains, impact on transcription control regulating chromatin modifications and RNA splicing[7–9]. However, whether they couple mechanosensing and the control of gene expression is not known. Phosphatidyl Inositol 5-Phosphate 4 Kinases (PI5P4K/PIP4K) are lipid kinases able to phosphorylate PtdIns5P to generate small pools of PtdIns(4,5)P2 compartmentalised at specific cell organelles[10,11]. Three isoforms of PIP4K exist in mammalian cells, PIP4K2A, 2B and 2C, mainly localizing in the cytoplasm, nucleus and endomembrane compartments respectively[12–15]. Deregulations of these lipid phosphotransferases have been reported in different human cancers[11,16,17].

Here we show that PIP4K2B responds to substrate mechanics, and that its depletion, impacting on UHRF1 expression, drives both

decrease in nuclear envelope tension and chromatin decompaction. This finally induces YAP cytoplasmic retention, rewiring of cytoskeletal organisation and impairment of cell motility.

## Results

### PIP4K2B protein levels positively correlate with substrate stiffness

Cytoplasmic and plasma membrane PIs and PI related enzymes play both direct and indirect roles in cell mechanics[18,19]. Here, we adopted hTERT_RPE1 cells as model of normal epithelial cells to investigate a possible role for PIP4K in mechanotransduction. Thus, we cultured cells onto fibronectin (FN)-coated acrylamide gels (2.3, 30 and 55 kPa) or onto glass coverslips coated with FN or poly D-lysine (PDL) (Fig. 1A).

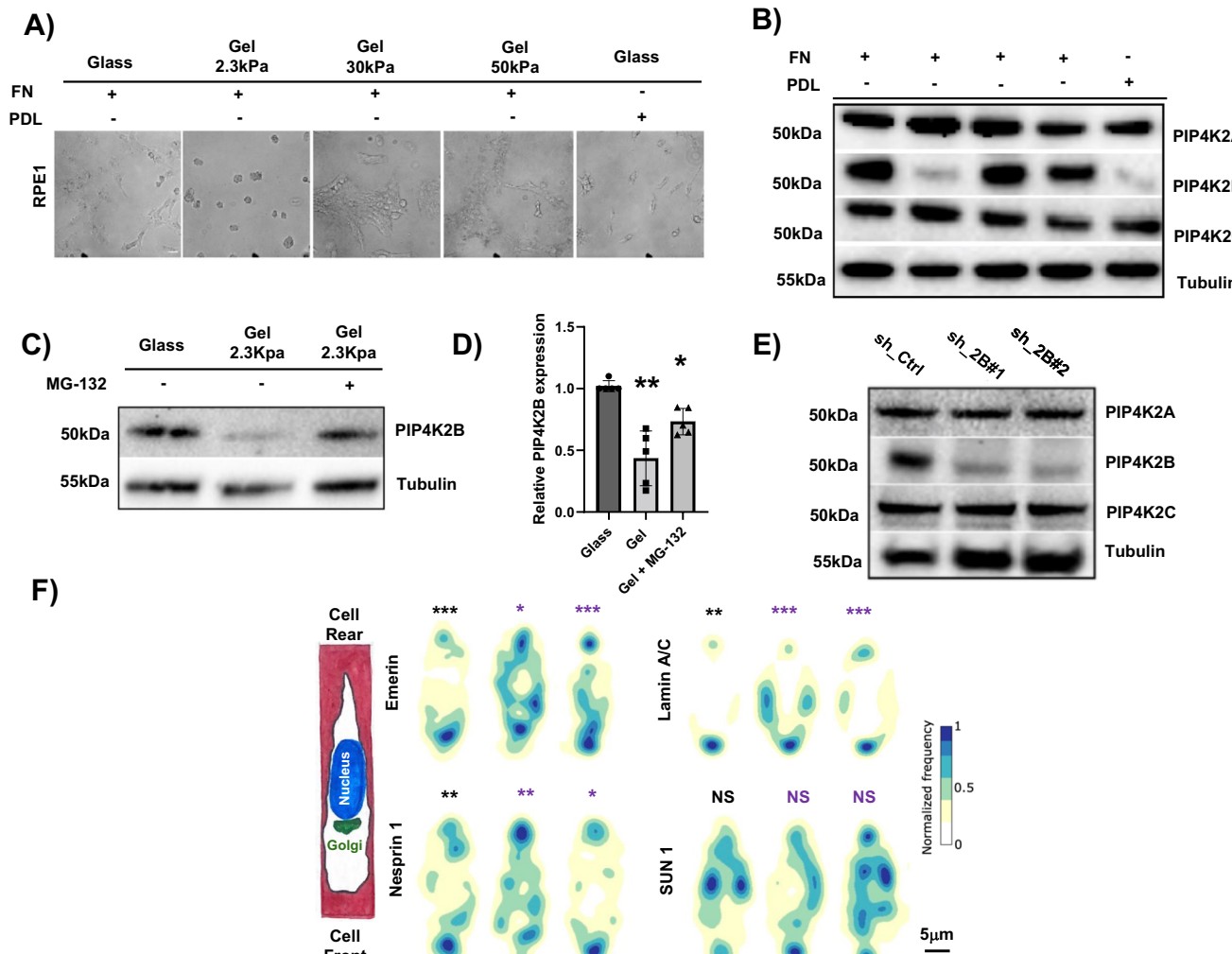

**Fig. 1 | PIP4K2B depletion impairs nuclear mechanical properties.**
**A** hTERT_RPE1 cells were cultured for 24 h on Fibronectin (FN)- or Poly D-lysine (PDL)-coated glass coverslips, and on FN-coated hydrogels of different stiffness (2.3 kPa / 30 kPa / 55 kPa) (scale bar = 20 μm). **B** Cell lysates from **A** were immunoblotted to analyse PIP4K2A, PIP4K2B and PIP4K2C protein levels. β-Tubulin was used as loading control. **C, D** hTERT_RPE1 cells were cultured as in **A**. MG-132 (5 mM) was added at cell culture medium for 16 h. Cells growing on Glass, Gel 2.3 kPa and Gel 2.3 kPa + MG-132 were lysed and cell lysates immunoblotted to analyse PIP4K2B protein levels. β-Tubulin was used as loading control. Western Blots were quantified as ratio between intensity of PIP4K2B lanes versus β-Tubulin lanes. Results are reported as average of $n = 5$ independent experiments and represented as bar charts +/− standard deviation (Gel pVal 0.035, Gel/MG-132 pVal 0.0368).
**E** hTERT_RPE1 cells were transduced with two different sh_RNAi targeting PIP4K2B (sh_2B#1 and sh_2B#2). Western Blotting analysis was performed to analyse the

levels of PIP4K2B together with PIP4K2A and PIP4K2C to exclude possible off-target effects of the silencing procedure on other isoforms. Empty pLKO_1 vector was used as Ctrl (sh_Ctrl). **F** Nuclear polarity analysis of cells seeded on PLL-g-Peg linear (10 μm) patterns: distribution maps of the signal intensity of nuclear proteins analysed through immunofluorescence (Emerin, $n = 49/51/46$, Lamin A/C, $n = 50/67/50$, Nesprin 1, $n = 41/42/54$ and SUN1, $n = 34/50/49$) are reported. Scale bar 5 μm. Black * indicates statistical analysis of the front vs rear accumulation, while purple * indicates statistical analysis of Ctrl cells vs PIP4K2B knock-down cells. Statistical analyses were performed on at least three independent experiments using unpaired two-tailed Student's $t$ test with Welch's correction for 1D, two-sided Kolmogorov–Smirnov to test front versus back or two-sided Cramer-von Mises to test sh_Ctrl versus sh_PIP4K2B conditions for maps distribution reported in 1F with $p$ values as $*p < 0.05$, $**p < 0.01$, $***p < 0.001$.

24 h after cell seeding, among the three PIP4K isoforms, only PIP4K2B expression resulted strongly decreased in cells seeded onto soft (2.3 kPa gels) or low adhesion mimicking substrates (PDL-coated coverslips) (Fig. 1B). Drop of PIP4K2B on soft substrates was confirmed in murine embryonic fibroblast (MEF) cells, and in three cancer cell lines: Hela, PC3 (prostate cancer) and MDA-MB-231 (Breast Cancer) (Supplementary Fig. 1A). Interestingly, RT-qPCR analysis showed that the reduction of PIP4K2B protein levels was not due to impaired RNA levels (Supplementary Fig. 1B). However, an over/night treatment of cells seeded on soft substrates (2.3 kPa gels) with the proteasome inhibitor MG-132 partially rescued PIP4K2B protein levels, suggesting protein degradation as the main cause of PIP4K2B drop (Fig. 1C). We next investigated how direct depletion of PIP4K2B, highly present in hTERT_RPE1 nuclei (Supplementary Fig. 1C), could mimic cell responses to soft substrates.

## PIP4K2B depletion impacts nuclear polarisation

We and others recently showed that cell polarity is partially transmitted to the nucleus through the polarisation of nuclear envelope (NE) proteins[20,21]. To investigate the role of PIP4K2B in nuclear polarity, we plated hTERT_RPE1 cells on FN-coated micro-patterned lines of 10 μm in width, as previously described[20]. This allowed the appearance of a clear front-rear polarity, as assessed by Golgi staining, used as marker of cell front (Supplementary Fig. 2A). We then silenced PIP4K2B (sh_2B#1 and sh_2B#2) (Fig. 1E) and analysed by immunofluorescence the distribution of NE components. Remarkably, we found that Lamin A/C, Emerin and Nesprin 1 were strongly delocalised upon PIP4K2B depletion (Fig. 1F and Supplementary Fig. 2B), while expression of NE components remained unchanged when compared with control cells (sh_Ctrl) (Supplementary Fig. 2C). In addition, exploiting a fluorescence probe (ING2-PHD_GFP)[22–24] to detect PtdIns5P, the lipid substrate of PIP4K2B, this was found enriched at the front of cell nucleus in control cells. (Supplementary Fig. 2D, E). This preferential localisation was lost in cells depleted for PIP4K2B, while the cytoplasmic counterpart was spared. Altogether, our findings suggest that PIP4K2B functions mainly in the nucleus and that its depletion impairs nuclear polarity.

## A FRET sensor based on Nesprin 1 shows altered NE tension in cells depleted for PIP4K2B

Nuclear envelope organisation regulates connections between nucleus and cytoplasm, translating forces from the cytosol to the nucleus and controlling NE tensional state[25]. In order to analyse if this could be affected by PIP4K2B depletion, we developed an ad hoc fluorescence energy transfer (FRET) sensor using the LINC complex protein Nesprin-1 as backbone, named Mini Nesprin 1, which exploited orientation-based FRET probes (cpst)[26]. Nesprins possess a Klarsicht, ANC-1, Syne Homology (KASH) domain at C-Terminus which allows their anchorage to NE through the link with SUN-domain proteins (SUN-proteins binding at perinuclear membrane), and two N-Terminus Calponin domains (CH-CH) for Actin binding[27]. Nesprins are then subjected to a certain degree of force depending on the cell mechanical state[28]. Mini Nesprin 1 showed clear localisation at NE and, as the wildtype protein, front polarisation (Supplementary Fig. 3A, B). Moreover, its expression did not alter cell viability or proliferation (Supplementary Fig. 3C–E). As control we also developed a mutant Mini Nesprin-1, lacking the CH-CH domain. Due to the impaired actin binding, this construct did not show significant changes in ratiometric FRET signals in cells seeded on either FN- or PDL-coated coverslips (Supplementary Fig. 3F–H). Lack of PIP4K2B induced a strong decrease in NE tension (Fig. 2A), similarly to what is observed in cells seeded on a soft substrate (Gel 2.3 kPa), on PDL-coated glass coverslips or treated with Latrunculin (Supplementary Fig. 3H).

## PIP4K2B depletion impacts chromatin compaction through alterations of heterochromatin

Alterations in nuclear mechanics are usually associated with changes in chromatin organisation[4,29]. To study the effects of lack of PIP4K2B on chromatin, we performed a chromatin compaction assay exploiting fluorescence energy transfer (FRET)-based fluorescent lifetime imaging microscopy (FLIM)[30]. More specifically, cells were transduced to knock-down PIP4K2B and subsequently co-transfected with histone H2B tagged with either -GFP (donor) or -mCherry (acceptor). Upon PIP4K2B depletion the fluorescent lifetime of H2B-GFP was increased, indicating a decrease in chromatin compaction (Fig. 2B). Strikingly, this chromatin status was similar to one of cells seeded on soft gels (2.3 kPa), reinforcing the idea that the loss of PIP4K2B is mimicking the cell response to soft substrate. In order to corroborate our findings, we analysed levels of Histone 3 lysine 9 trimethylation (H3K9me3), a well-established marker of heterochromatin, which resulted decreased in cells depleted for PIP4K2B, as well as in cells seeded on a soft 2.3 kPa gel (Fig. 2C)[31]. Next, Transmission Electron Microscopy (TEM) analysis also confirmed that heterochromatin was strongly altered in PIP4K2B knock-down cells (Fig. 2D and Supplementary Fig. 4A, whole images). Interestingly, cells depleted for PIP4K2B also presented nuclear invaginations in line with low NE tensional state (Fig. 2D and Supplementary Fig. 4A–C). These data suggest that mechano-induced downregulation of PIP4K2B drives changes in chromatin organisation.

## PIP4K2B positively regulates YAP nuclear/cytoplasmic ratio

To further test how PIP4K2B depletion mimic cell response to soft, we investigated whether this could also affect the distribution of the master regulator of mechanotransduction processes YAP[32]. We found that strong nuclear localisation of YAP was present in control cells seeded on FN-coated glasses, while nuclear/cytoplasmic ratio was deeply impaired in cells depleted for PIP4K2B (Fig. 3A, B). Consistently, immunoblot analyses showed accumulation of YAP phosphorylation at Ser-127, a well-recognised marker of cytoplasmic retention of YAP (Fig. 3C)[33,34]. Altogether these observations show that lack of PIP4K2B impacts fundamental nuclear mechanical properties, as NE tension and chromatin compaction, and it also impairs correct nuclear/cytoplasmic shuttling of YAP.

## RNA sequencing unravels PIP4K2B role in mechanotransduction and chromatin remodelling

Since depletion of PIP4K2B impairs YAP nuclear localisation, we performed RNA-sequencing analysis to investigate how this could affect the transcriptomic landscape of the cells. hTERT_RPE1 cells seeded on FN-coated glass coverslips were then transduced to knock-down PIP4K2B with three different sh_RNAi (sh_#1/#2/#3, 1st condition as sh_PIP4K2B), and with empty pLKO_1 vector as controls (2nd condition as Control). In addition, cells transduced with empty pLKO_1 vector and seeded on 2.3 kPa gels were used as soft-seeded cells (3rd condition as Soft). Interestingly, in line with data showed above, RNA-seq analysis indicated genes from a YAP conserved signature as more highly enriched in control cells compared to either depletion of sh_PIP4K2B or seeding cells on a soft substrate (Cordenonsi_YAP_Conserved_Signature). Some of the DEGenes were validated through RT-qPCR (Supplementary Fig. 5A). This strengthened the concept that lack of PIP4K2B could mimic cell response to soft substrate also at the transcriptomic level (Fig. 3D and Supplementary Fig. 5A). In particular, once we analysed DEGenes of sh_PIP4K2B vs Control and Soft vs Control, we found an important degree of overlap of genes regulated by PIP4K2B depletion and soft-seeding (Fig. 3D). Many DEGenes shared by Soft and sh_PIP4K2B were involved in processes like chromatin remodelling and histone methylation/demethylation (Fig. 3D). Moreover, once GO terms were investigated, we found a strongly negative correlation of genes involved in heterochromatin formation in sh_PIP4K2B (Supplementary Fig. 5B). Nuclear softening

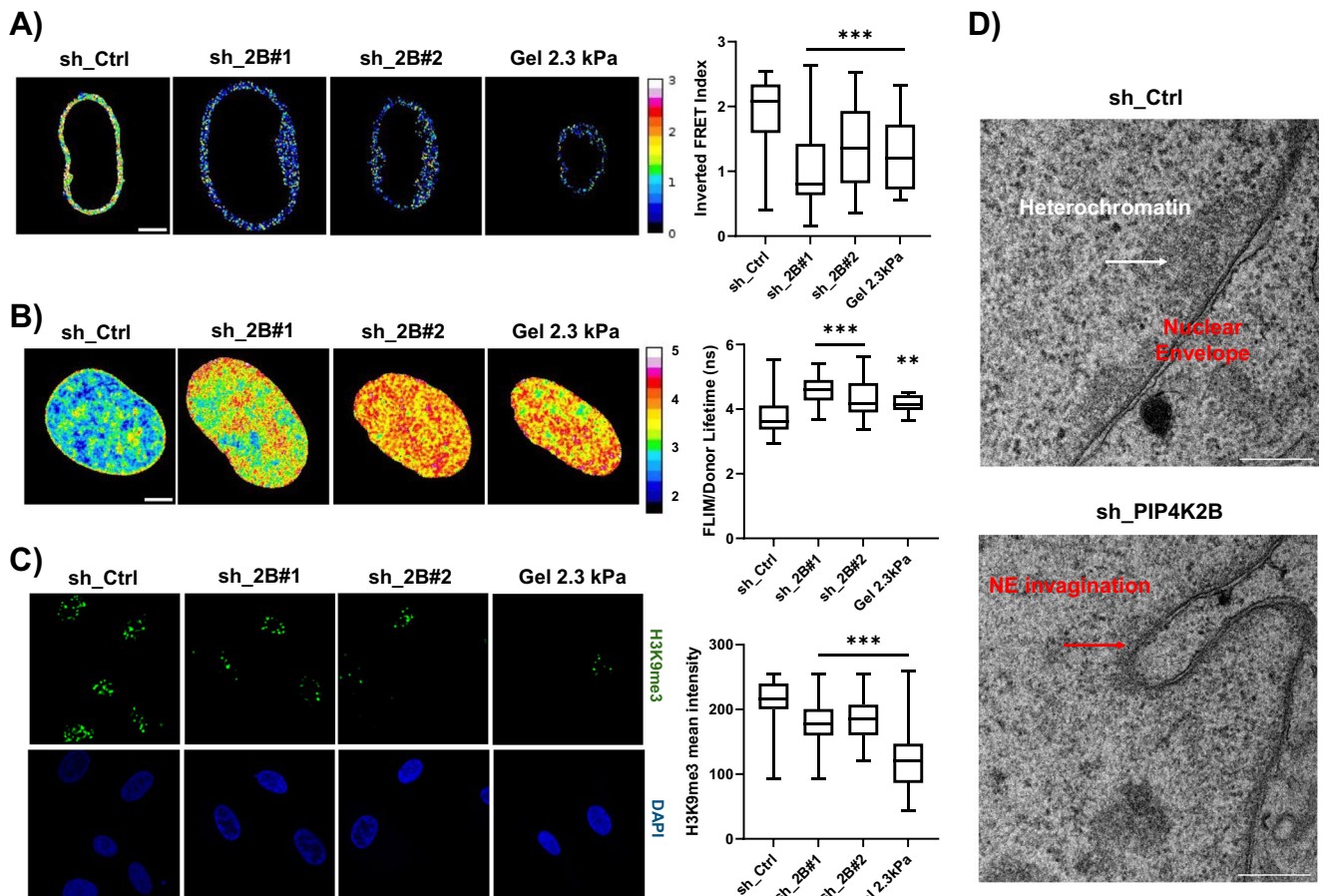

**Fig. 2 | PIP4K2B impacts on chromatin organisation altering heterochromatin levels and drives YAP cytoplasmic localisation. A** Nuclear envelope (NE) tension analysis exploiting Mini Nesprin 1 cpst-FRET sensor. Representative nuclei of Ctrl cells (sh_Ctrl, n = 47) and cells depleted for PIP4K2B (sh_2B#1/#2, n = 51/49) are shown. As internal control, cells seeded on 2.3 kPa gels were used (n = 30) (scale bar = 20 μm). All coverslips were coated with Fibronectin prior usage. Data quantification is shown as boxplot chart representing inverted FRET index values (donor/acceptor, the higher the value, the higher the tension) and plotted as Min to Max in Prism9 software. Data derive from n = 3 independent experiments. **B** FLIM-FRET chromatin compaction assay performed transfecting cells with H2B-GFP and H2B-mCherry tagged plasmids. Sample images of Fluorescent lifetime (FLIM-FRET index) of cells transduced with pLKO_1 Ctrl (sh_Ctrl, n = 39) and pLKO_1 sh_PIP4K2B (sh_2B#1/#2, n = 29/24). As internal control, cells seeded on 2.3 kPa gels were used (n = 12, pVal 0.032). All coverslips were coated with Fibronectin prior usage. Data are represented as boxplots and overall Lifetime is shown (higher value, lower compaction) (scale bar = 20 μm). **C** H3K9me3 staining of nuclei of cells transduced with pLKO_1 Ctrl (sh_Ctrl) and pLKO_1 sh_PIP4K2B (sh_2B#1/#2). As internal control, cells seeded on 2.3 kPa gels were used. Data quantification is shown as boxplot representing the ratio between the intensity of single fluorescent dots and cell nuclear area (scale bar = 20 μm). Data derived from n = 3 independent experiments. **D** Transmission electron microscopy (TEM) derived images from routine 60 nm EM section. Coloured arrowheads indicate heterochromatin (white), nuclear envelope (NE, black), and NE invaginations (green) respectively (scale bar 1000 ηm). Data are representative of n = 2 experiments. Statistical analyses were performed using unpaired two-tailed Student's t test with Welch's correction with p values as *p < 0.05, **p < 0.01, ***p < 0.001. In boxplots: middle bars are medians, the rectangles span from the first to the third quartiles and bars extent from min to max values.

can be driven by decreased levels of heterochromatin[35]. In line with this, we found several deregulated genes involved in chromatin remodelling in both Soft and sh_PIP4K2B conditions. As presented in the Volcano plots and heatmap shown in Fig. 3E and Supplementary Fig. 5C, D, shared DEGenes sets could be divided in three different clusters: (1) genes down in Ctrl vs sh_PIP4K2B/Soft (blue); (2) genes up in Ctrl vs sh_PIP4K2B/Soft (red); (3) genes differentially regulated in Soft compared to sh_PIP4K2B (green). We then focused our attention on two genes whose expression was decreased in both sh_PIP4K2B and Soft (cluster 3): SUV39H1 and UHRF1 (Fig. 3E and Supplementary Fig. 5D, E). The first encodes a histone methyltransferase which methylates H3K9me2 into H3K9me3 allowing heterochromatin formation[36]. The second encodes for a multidomain nuclear E3-ubiquitin ligase involved in heterochromatin organisation, taking part in H3K9 binding of SUV39H1[36], and in the recruitment of DNA methyltransferases to control DNA methylation[37]. Interestingly UHRF1 protein decrease was found in PIP4K2B depleted cells as well as in cells

seeded on soft substrates, while SUV39H1 protein levels were not altered (Supplementary Fig. 6A, B). We recently proposed UHRF1 as a possible target of PIP4K2B signalling, while others showed that its function is tightly connected to its binding to PtdIns5P[38,39]. In addition, UHRF1 gene has been identified as a target of YAP/TEAD transcription factors[40]. Then, we decided to investigate how PIP4K2B could be involved in UHRF1 control, and if this was mediated by YAP function.

## PIP4K2B depletion impacts on the chromatin remodeller UHRF1 at post-transcriptional level
Our data showed that PIP4K2B silencing drives nuclear envelope tension drop, heterochromatin remodelling and transcriptomic rewiring, possibly due to YAP cytoplasmic retention and UHRF1 depletion. Then, to understand the role of YAP and UHRF1 in the altered nuclear mechanics driven by PIP4K2B, we investigated in time-dependent manner their evolution upon PIP4K2B KD for 4 days. Interestingly, we found that knock-down of PIP4K2B led to

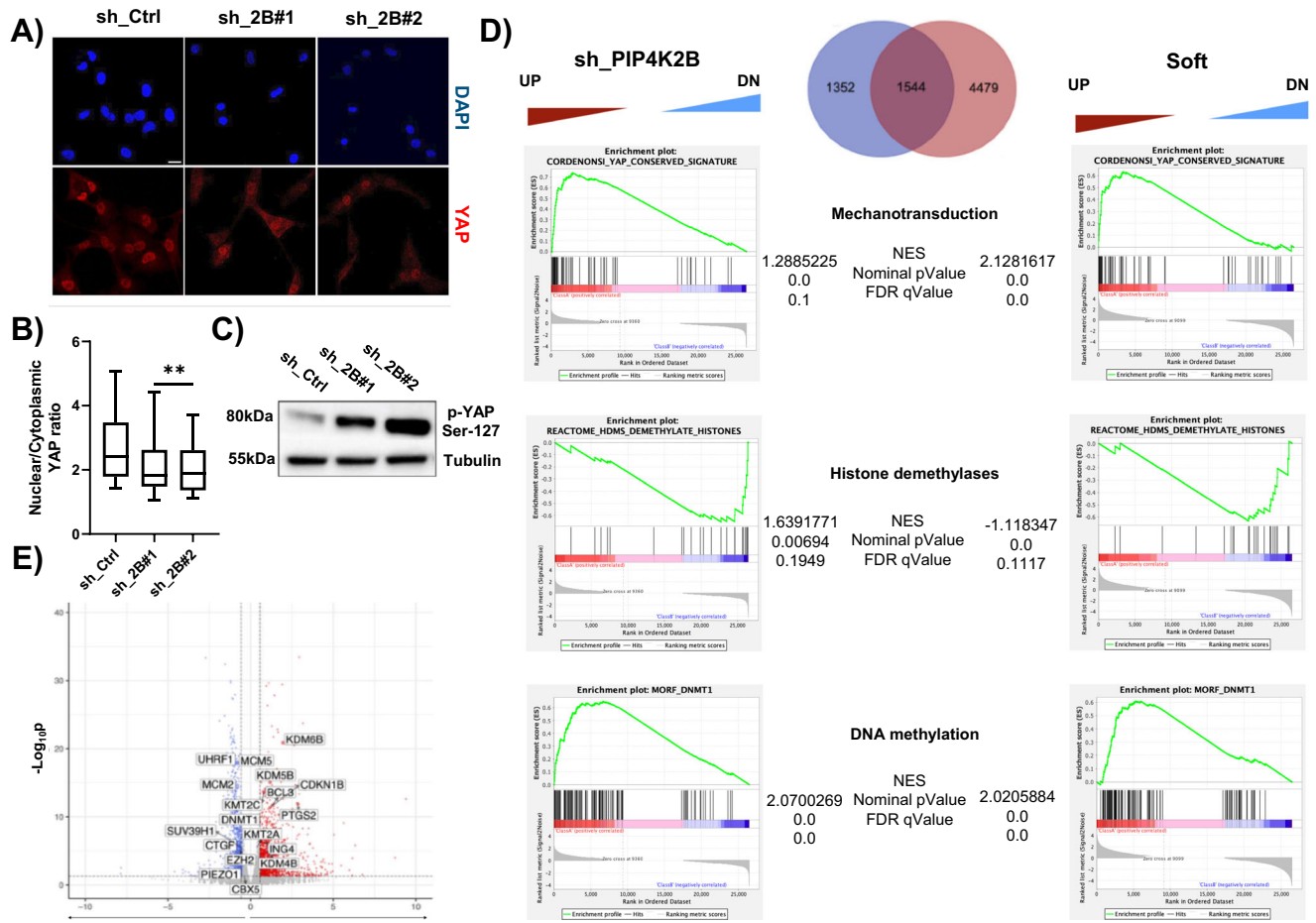

**Fig. 3 | PIP4K2B depletion rewires hTERT_RPE1 cells transcriptomic profile through alteration in mechanotransduction and chromatin remodelling genes. A** Immunofluorescent staining of YAP cellular distribution in sh_Ctrl and sh_2B#1/#2 cells. **B** Quantification of nuclear to cytoplasmic YAP signal ratio are reported (scale bar = 20 μm). Data derived from $n = 3$ independent experiments. In boxplots: middle bars are medians, the rectangles span from the first to the third quartiles and bars extent from min to max values. Statistical analysis was performed using unpaired two-tailed student's $t$ test with Welch's correction (sh_2B#1 pVal 0.013, sh_2B#2 pVal 0.022). **C** Western Blotting analysis was performed to analyse the levels of phosphorylated-YAP (Ser-127) in sh_Ctrl and sh_2B#1/#2 cells. **D** Gene expression data generated through RNA-seq comparing Ctrl vs sh_PIP4K2B and Ctrl vs Soft were used for GSEA analyses to extract biological knowledge. Venn-Diagram show number of DEGenes in sh_PIP4K2B and Soft conditions if compared to Ctrl.

GSEA enrichment plots of differentially expressed genes using the Cordenonsi YAP conserved signature (M2871), Reactome_HDMS_Demethylate_Histones (R-HSA-3214842) and MORF-DNMT1 (M10146). The green curve in the charts corresponds to the enrichment score (ES) curve, which is the running sum of the weighted ES obtained from GSEA software, while the normalised enrichment score (NES), the corresponding one-sided $p$ value and the false discovery rate (FDR, Benjamini and Hochberg method) are reported within the graph. **E** Volcano plot representing DEGenes in sh_PIP4K2B vs Ctrl, with highlight of genes involved in chromatin modifications. All significantly ($p$Adj < 0.01, wald test $p$ value adjusted using Benjamini and Hochberg method) deregulated genes are in red (upregulated) and blue (downregulated). Enrichment (log2(fold change)) is plotted on the $x$-axis and significance (Wald test −log10($p$-value two-sided)) is plotted on the $y$-axis.

UHRF1 levels drop prior to changes in YAP phosphorylation (Fig. 4A). This was confirmed by immunofluorescence staining which showed YAP cytoplasmic retention only at 72 h and not at 48 h (Supplementary Fig. 6C). These data suggest that PIP4K2B controls UHRF1 independently by YAP nuclear localisation. To further explore if PIP4K2B activities require YAP, we expressed a nuclear mutant of YAP (YAP-S6A) in hTERT_RPE1 cells depleted for PIP4K2B. While in control cells, as expected, YAP-S6A induced a significative increase of UHRF1 protein level, in PIP4K2B depleted ones, its expression was not sufficient to rescue UHRF1 levels (Fig. 4B). Altogether these results suggest a post-transcriptional effect of PIP4K2B on UHRF1. They also show that YAP transcriptional activity is not necessary or sufficient to rescue the strong alterations driven by lack of either PIP4K2B or UHRF1. PIP4K depletion usually increases levels of its substrate, PtdIns5P, controlling different nuclear outputs including UHRF1 capacity to bind H3K9me3[39,41]. To mimic the depletion of PIP4K2B, hTERT_RPE1 cells

were incubated with exogenous PtdIns5P (+ PI5P). Strikingly, we found that UHRF1 levels started to decrease just after 4 h of PtdIns5P treatment, while YAP was still located mainly in the nuclei of the cells (Fig. 4C, D). In addition, H3K9me3 levels decreased upon PtdIns5P treatment (Supplementary Fig. 6D). This confirms that PIP4K2B impacts primarily on UHRF1 and, subsequently, on YAP signalling. Considering that exogenous PtdIns5P triggered UHRF1 downregulation within 4 h, we investigated if this could be due to a post-transcriptional regulation of the protein. We then over-expressed UHRF1 in cells previously transduced with sh_Ctrl or sh_PIP4K2BWe found that also exogenous UHRF1 levels were decreased in cells lacking PIP4K2B, while MG-132 treatment could rescue UHRF1 protein in the cells (Supplementary Fig. 6E, F). Moreover, this was mirrored also in cells seeded in soft-like (PDL) substrates (Supplementary Fig. 6G). These data indicated a strong degradation rate of UHRF1 in cells depleted for PIP4K2B, which was confirmed by Ubiquitination assay (Fig. 4E).

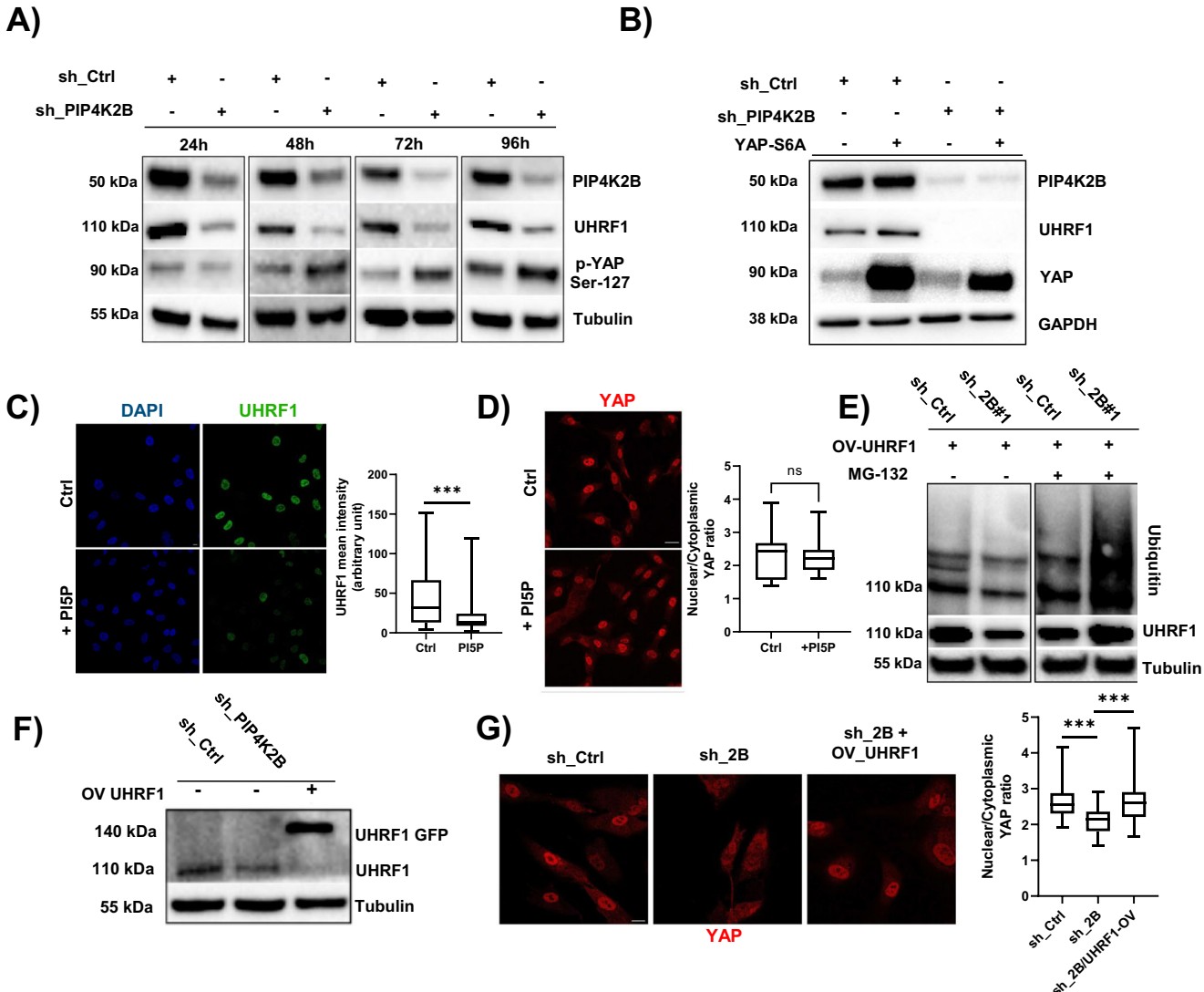

**Fig. 4 | PIP4K2B/UHRF1 signalling controls YAP intracellular localisation.**
**A** Time course of the expression of the levels of UHRF1 and phosphorylated-YAP (Ser-127) was assessed by western blotting. WB was performed 48 h/72 h/96 h/120 h after that hTERT_RPE1 cells were transduced with sh_Ctrl or sh_2B#1/#2.
**B** hTERT_RPE1 cells were transduced with sh_2B or sh_Ctrl and 48 h later, with Empty vector or YAP-S6A in order to express a nuclear mutant of YAP. UHRF1 levels were analysed by western blotting. **C** Immunofluorescent staining of UHRF1. Cells were starved for 16 h, then seeded in complete medium supplemented with PtdIns5P (+PI5P). As control, cells were starved and grown in complete medium (Ctrl) (scale bar = 20 μm). Quantification of the mean intensity of UHRF1 is reported. Data are representative of $n = 3$ independent experiments. **D** Cells were treated as in **C** and immunofluorescent staining of YAP cellular distribution was performed (scale bar = 20 μm). Quantification of nuclear to cytoplasmic YAP signal ratio is reported. Data are representative of $n = 3$ independent experiments.
**E** Ubiquitination assay showing degradation of UHRF1 in cells transduced to silence PIP4K2B (sh_2B#1/#2). Cells overexpressing UHRF1 were treated with MG-132

(5 mM) for 16 h. As control, cells growing in complete medium were used. UHRF1 was then immunoprecipitated, and western blotting analysis was performed to analyse the levels of Ubiquitinated-UHRF1 (Ub IB). 5% of input samples were used to assess amount of UHRF1 immunoprecipitated, while cell lysates were immuno-blotted for Tubulin to check loading conditions. **F** Western Blotting analysis of UHRF1 protein expression in cells depleted for PIP4K2B (sh_2B) and concomitantly depleted for PIP4K2B and transduced to overexpress GFP-UHRF1. As control, cells transduced with pLKO_1 vector were used (sh_Ctrl). **G** Immunofluorescent staining of YAP localisation in cells depleted for PIP4K2B (sh_2B), depleted for PIP4K2B and concomitant overexpression UHRF1 (sh_2B / OV_UHRF1), and control cells (sh_Ctrl) (scale bar = 20 μm). Quantification of nuclear to cytoplasmic YAP signal ratio is reported Data are representative of $n = 3$ independent experiments. In boxplots: middle bars are medians, the rectangles span from the first to the third quartiles and bars extent from min to max values. Statistical analyses were performed using unpaired two-tailed Student's $t$ test with Welch's correction with pValues as ***$p < 0.001$. At least 50 cells were analysed for every IF experiment.

## UHRF1 depletion phenocopies YAP cytoplasmic retention encountered in cells lacking PIP4K2B

We next investigated if UHRF1 depletion could phenocopy PIP4K2B depletion. We thus directly silenced UHRF1 with two different sh_RNAs (sh_UHRF1#1/#2). UHRF1 knock-down drove YAP cytoplasmic retention (Supplementary Fig. 6H, I). Strikingly, despite the high-rate of UHRF1 degradation in PIP4K2B knock-down cells, once re-expressed it in these cells, this rescued nuclear YAP localisation (Fig. 4F, G and Supplementary Fig. 6J).

These data clearly indicate that PIP4K2B controls YAP through UHRF1.

## PIP4K2B/UHRF1 axis impacts on cytoskeleton and plasma membrane organisation

YAP signalling modulates cell mechanics impacting cell shape, actin cytoskeleton and plasma membrane organisation[42,43]. In accordance, cells depleted for PIP4K2B or seeded on soft surfaces (gel 2.3 kPa) were characterised by smaller cell area, defects in actin-network, decreased

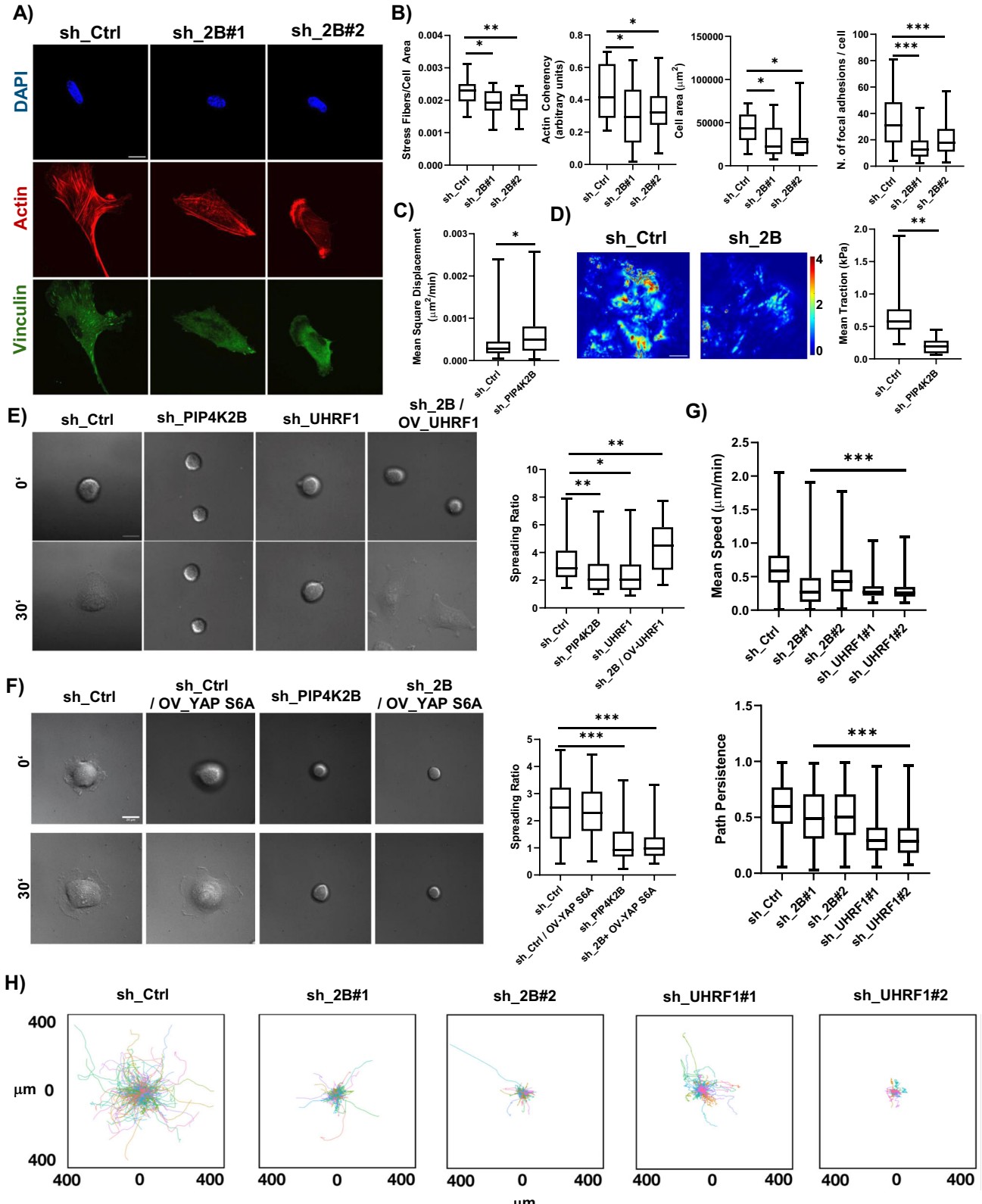

number of stress fibres and focal adhesions (Fig. 5A, B and Supplementary Fig. 7A). This cytoskeletal rewiring was mirrored by decreased phosphorylation levels of focal adhesion kinases (p-FAK) and Myosin Light Chain 2 (p-MLC2) in cells lacking PIP4K2B (Supplementary Fig. 7B). Moreover, direct knock-down of UHRF1 phenocopied PIP4K2B depletion in terms of impaired number of focal adhesions (Supplementary Fig. 7C). Interestingly, defects in cytoskeletal organisation

induced by PIP4K2B KD led to altered cytoplasmic rheology and cell capacity to exert forces on the substrate (Fig. 5C, D). We then analysed the temporal dynamics of Focal Adhesion Kinase (FAK) phosphorylation, a marker of FA assembly, by western blotting in cells silenced for PIP4K2B, where UHRF1 levels and YAP phosphorylation have been previously assessed (See Fig. 4A). Interestingly, we found that phosphorylation of FAK was lost after 96 h of PIP4K2B silencing, an event

**Fig. 5 | PIP4K2B/UHRF1 signalling controls cytoplasmic organisation, impacting on cell capacity to spread and move. A** Immunofluorescent staining of Actin (Phalloidin), Vinculin (Focal Adhesions) and nuclei (DAPI) of cells depleted for PIP4K2B (sh_2B#1/#2). Control cells were transduced with empty pLKO_1 vector (sh_Ctrl) localisation. **B** Analysis of actin (sh_2B#1 pVal 0.0187, sh_2B#2 pVal 0.071), focal adhesion number (sh_2B#1 and sh_2B#2 pVal <0.001), cell area (sh_2B#1 pVal 0.0119, sh_2B#2 pVal 0.0178) and actin coherency (sh_2B#1 pVal 0.0365, sh_2B#2 pVal 0.0474). sh_Ctrl $n = 18$, sh_2B#1 $n = 25$, sh_2B#2 $n = 25$ (scale bar = 10 µm). **C** Cell Rheology analysis, performed measuring the mean square displacement of microparticles injected in the cell cytoplasm (sh_Ctrl $n = 40$, sh_2B $n = 40$, pVal 0.0108). **D** Traction Force Microscopy (TFM) analysis performed on cells seeded for 24 h on fibronectin-coated silicone (elastic modulus of 5.16 kPa) samples containing QDs. 14 cells were analysed for sh_Ctrl, 25 for sh_PIP4K2B, pVal 0.0012. Scale bar 30 mM. **E** Representative images of cell spreading performed on cells seeded on fibronectin-coated glass coverslips taken at Time 0' (cell attachment to the substrate) and at Time 30' (cells in active spreading) (scale bar = 20 µm). Cells were transduced to silence PIP4K2B (sh_PIP4K2B, $n = 55$, pVal 0.0057) or UHRF1 (sh_UHRF1, $n = 30$, pVal 0.0186), and to concomitantly overexpress UHRF1 and silence PIP4K2B (sh_2B/OV_UHRF1, $n = 27$, pVal 0.0088). Control cells were transduced with empty pLKO_1 vector (sh_Ctrl, $n = 35$). **F** Cell spreading assay performed as in **E**. Cells were transduced to silence PIP4K2B (sh_PIP4K2B) or with an empty control vector (sh_Ctrl). Cells were transduced 48 h later to express nuclear YAP-S6A (sh_2B / OV_YAP-S6A and sh_Ctrl /OV_YAP-S6A, pVal <0.001). More than 50 cells per conditions were analysed. **G, H** 2D cell motility analysis of cells seeded on fibronectin-coated glass coverslips. **G** Data quantification of mean speed and path persistence of the cells. **H** Plots representing trajectories of cells depleted for PIP4K2B or UHRF1, and Control cells. For **G** and **H** data are representative of $n = 3$ independent experiments. In boxplots: middle bars are medians, the rectangles span from the first to the third quartiles and bars extent from min to max values. Statistical analyses were performed using paired two-tailed Student's $t$ test, with $p$ values as *$p < 0.05$, **$p < 0.01$, ***$p < 0.001$.

that occurred only after loss of UHRF1 and YAP phosphorylation, respectively at 48 and 72 h (Supplementary Fig. 7D, complete WB is presented). In line with our findings, IF analyses showed impairment of FA and actin cytoskeleton organisation in cells depleted of PIP4K2B only after 96 h (Supplementary Fig. 7E, F). Moreover, YAP-S6A expression in PIP4K2B depleted hTERT-RPE1 cells did not rescue focal adhesion (FA) number (Supplementary Fig. 7G). Altogether, these data indicate a hierarchical model through which PIP4K2B primarily impacts on UHRF1 and, next, cell mechanics.

**PIP4K2B/UHRF1 signalling controls cells spreading and motility**
Since depletion of PIP4K2B or UHRF1 deeply impacted cell mechanics, we investigated how this could be reflected in the motile ability of the cells. We first performed cell spreading analysis[44]. Cells transduced to silence PIP4K2B or UHRF1 were allowed to spread on FN-coated glass coverslips. Cells depleted of PIP4K2B or UHRF1 had impaired spreading capacity if compared to control cells (Fig. 5E and Supp. Movie 1). Strikingly, once UHRF1 was re-expressed into cells lacking PIP4K2B, these cells could completely rescue their spreading ability (Fig. 5E and Supp. Movie 1). On the contrary, YAP-S6A expression in PIP4K2B depleted hTERT-RPE1 cells was not sufficient to recover it (Fig. 5F and Supp. Movie 2). In addition, once depleted for PIP4K2B, PC3 and MDA-MB-231 cells showed as well altered focal adhesion number and capacity to spread (Supplementary Fig. 8A–C). We then performed 2D motility assay to analyse cells capacity to move. Cells lacking PIP4K2B (with four different sh_RNAi, Supp. Movie 3) or UHRF1 (with two different sh_RNAi, Supp. Movie 4) were characterised by altered motility compared to control cells (Fig. 5G, H, Supplementary Fig. 9A–C).

**PIP4K inhibition through small molecular compounds mimics effects of PIP4K2B silencing**
Due to the increasing interest in the role of PIP4K signalling in cancer, several inhibitors have been developed. We decided to exploit recently synthesised compounds named A-131 and THZ-P1_2 to understand if pharmacological inhibition of these lipid kinases could recapitulate the depletion of PIP4K2B by sh_RNAi[45,46]. Although these compounds are not specific for PIP4K2B but inhibit different PIP4K enzymes, treating of hTERT_RPE1 with them led to impairment of UHRF1 levels, decreased NE tension and chromatin decompaction (Fig. 6A–C). This was followed by YAP cytoplasmic retention in treated cells (Fig. 6), and by a strong decrease in cell capacity to move (Fig. 6, Supplementary Fig. 9 and Supp. Movie 5).

## Discussion
Our study showed that the lipid kinase PIP4K2B functions as a substrate stiffness-related mechanoresponsive protein and controls nuclear mechanical properties, impacting on NE and chromatin state. Moreover, we elucidated a mechanism through which PIP4K2B

signalling alters levels of the chromatin remodeller UHRF1, inducing nuclear extrusion of YAP, cytoskeleton rewiring and impairment of cell motility.

Cells react to substrate stiffness and timely adapt to the mechanical properties of the microenvironment[47–49]. Here, we showed that cells growing on soft or low adhesion mimicking substrates (Gel 2.3 kPa and PDL-coated glass) had a strong and specific decrease in protein but not mRNA levels of PIP4K2B. Moreover, inhibiting proteasome-mediated degradation using MG-132 partially rescued PIP4K2B protein levels, suggesting a role for proteasome-mediated degradation of PIP4K2B in response to seeding on soft substrates. PIP4K2B mainly localises in the nucleus, where it is known to affect different nuclear activities, including chromatin remodelling and protein-histone binding, through the control of two lipid messengers PtdIns5P and PtdIns(4,5)P2[7,9]. PIP4K2B gene silencing well phenocopy its degradation on soft substrates. It induced indeed a decrease in FA number, smaller nuclei with clear nuclear invaginations, decreased nuclear envelope tension, decreased chromatin compaction and severe alterations of nuclear polarity.

Moreover, RNA-seq analysis confirmed the similarity between cells seeded on soft surfaces and PIP4K2B knock-down. These results highlighted the unexpected role of PIP4K2B/PtdIns5P signalling in the control of epigenetic and transcriptional outputs in response to mechanical stimuli. Interestingly, we found several genes involved in heterochromatin formation and maintenance as similarly expressed in soft-seeded and in PIP4K2B knock-down cells. This observation is consistent with a seminal paper reporting that, upon mechanical stress, alterations in heterochromatin, as induced by changes in levels of SUV39H1, the histone methyltransferases mainly responsible for H3K9me2 methylation into H3K9me3, could drive nuclear softening[35].

Surprisingly, while SUV39H1 mRNA levels were low both in soft-seeded and in PIP4K2B knock-down cells, its protein levels were not altered. This discrepancy could be due to an increased SUV39H1 protein stability as a consequence of oxidative stress, which is usually induced by soft substrates or by the depletion of PIP4K[50]. Similarly, UHRF1, a multi-domain protein involved in the maintenance of DNA methylation and in chromatin remodelling[37,51], was downregulated in both, soft-seeded and PIP4K2B knock-down cells. However, in this case, mRNA expression and protein amount were coherent. As part of its functions, UHRF1 acts as a scaffold protein for SUV39H1, facilitating its binding to H3K9me3[52]. Cells lacking UHRF1 are characterised by defects in DNA methylation, and altered heterochromatin conformation[37,51,53]. UHRF1 contains a Polybasic Region (PBR) able to bind PtdIns5P[39]. The binding with this lipid second messenger allosterically regulates UHRF1 protein folding and its histone binding capability. In addition, we recently found that UHRF1 is involved in PIP4K2B signalling, and others also showed that UHRF1 gene is a target of YAP[38,40]. Our findings suggest that PIP4K2B lack drives UHRF1

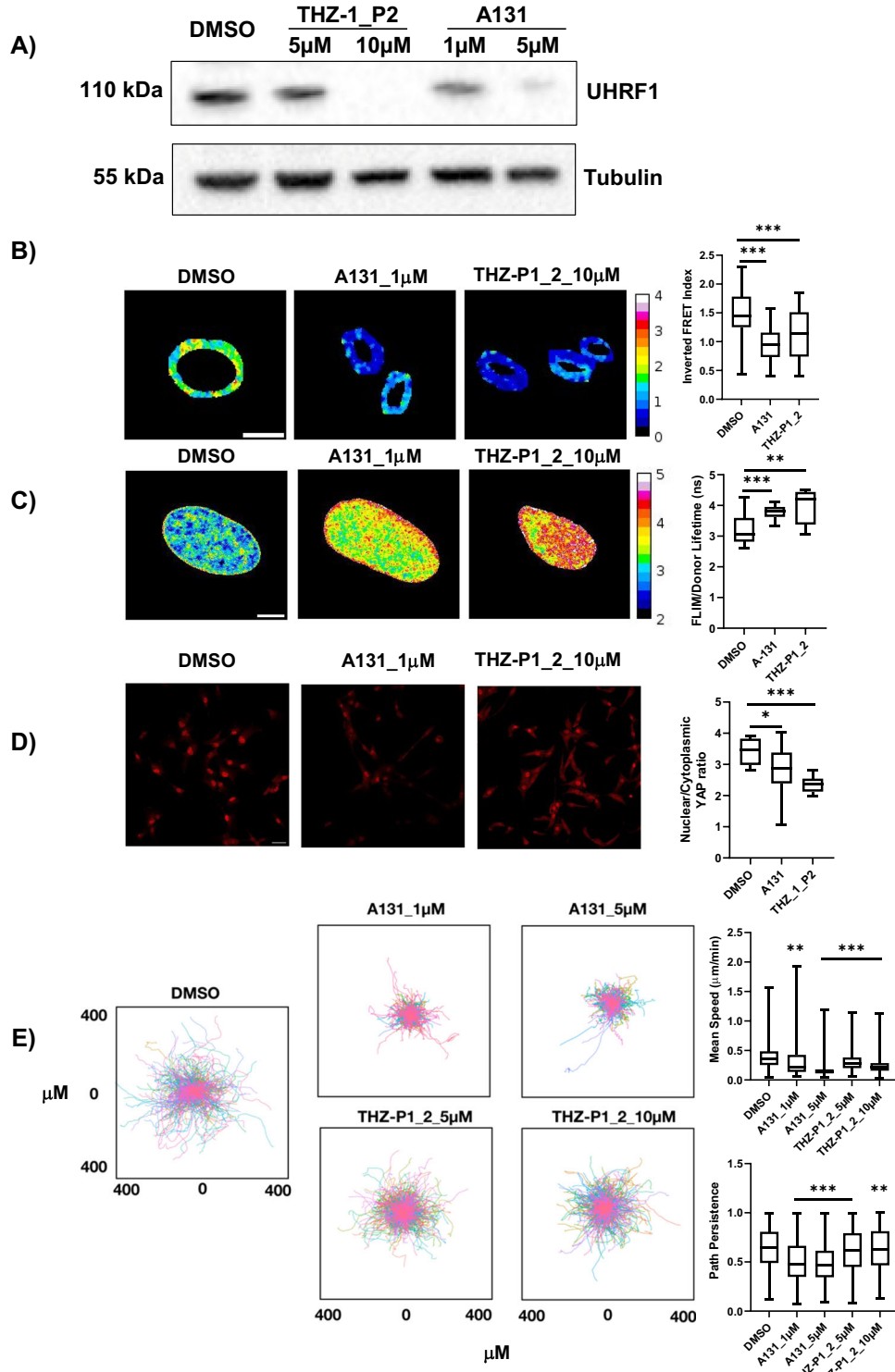

**Fig. 6 | Pharmacological inhibition of PIP4K signalling phenocopies cell soft-ening and defects in motility. A** Western Blot to assess UHRF1 expression in hTERT_RPE1 cells treated with PIP4K inhibitors A131 (1–5 mM) or THZ_P1_2 (5–10 mM). **B** FRET analysis of NE tensional state exploiting Mini Nesprin 1 cpst FRET sensor. Cells were treated with PIP4K inhibitors A131 (1 μM, $n = 37$, pVal <0.001) and THZ-P1_2 (10 μM, $n = 43$, pVal <0.001) for 24 h. DMSO was used as vehicle control (DMSO, $n = 41$) (scale bar = 10 μm). Data quantification is shown as boxplot chart representing inverted FRET index values (donor/acceptor, the higher the value, the higher the tension). **C** FLIM-FRET chromatin compaction assay per-formed transfecting cells with H2B-GFP and H2B-mCherry tagged plasmids (scale bar = 10 μm). Sample images of Fluorescent lifetime (FLIM-FRET index) of cells treated with PIP4K inhibitors as in **B**. Data are represented as boxplots and overall

Lifetime is shown (higher value, lower compaction). PIP4K inhibitor A131 (1 μM, pVal 0.0009) $n = 9$, THZ-P1_2 (10 μM, pVal 0.0034) $n = 9$, DMSO, $n = 14$. **D** Immunofluorescent staining of YAP localisation in cells treated as in **A**. Quanti-fication of nuclear to cytoplasmic YAP signal ratio is reported (scale bar = 20 μm), A131 (1 μM, pVal 0.0407) and THZ-P1_2 (10 μM, pVal <0.0001). **E** 2D cell motility analysis of cells seeded on fibronectin-coated glass coverslips. Data quantification of trajectories and mean speed/path persistence is shown. In boxplots: middle bars are medians, the rectangles span from the first to the third quartiles and bars extent from min to max values. Statistical analyses were performed using paired two-tailed Student's $t$ test, with $p$ values as *$p < 0.05$, **$p < 0.01$, ***$p < 0.001$. Experiments were repeated $n = 3$ times.

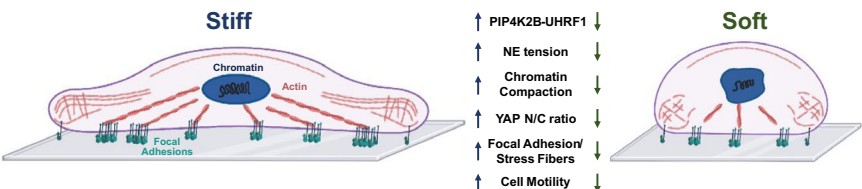

**Fig. 7 | Graphical summary.** Graphical description of mechanical properties of cells in function of PIP4K2B and UHRF1 expression. Actin fibres are represented in red, chromatin in black, nucleus in blue and focal adhesion complexes (integrins) in green. This figure was created with BioRender.com.

depletion. How this happens remains to be elucidated. Since previous works suggested that PtdIns5P accumulation could drive to enhanced ubiquitylation of SPOP (speckle-type POZ domain protein)−Cullin3 (Cul3, E3 ubiquitin ligase) substrates[8], it is tempting to speculate that depletion of PIP4K2B may drive UHRF1 degradation through this pathway. On the other hand, our data clearly indicate that this mechanism works independently by YAP function as co-transcription factor since expression of a nuclear mutant of YAP in cells depleted of PIP4K2B did not reverse UHRF1 decrease. Moreover, we showed that UHRF1 drop in cells silenced for PIP4K2B occurs prior than YAP phosphorylation and nuclear extrusion. Remarkably, direct silencing of UHRF1 phenocopied PIP4K2B knock-down in terms of YAP cytoplasmic accumulation, and UHRF1 overexpression in cells depleted for PIP4K2B could restore YAP localisation in the nuclei. These data suggest a hierarchical series of events which lead to impairment of YAP signalling: first, decreased PIP4K2B levels drive UHRF1 degradation; next, lack of PIP4K2B/UHRF1, inducing chromatin remodelling, impacts on YAP localisation; finally, YAP nuclear extrusion is in turn possibly responsible for the subsequent decreased mRNA levels of UHRF1 (Fig. 7).

Chromatin condensation is necessary for cell motility[54], while YAP signalling controls the expression of genes involved in focal adhesion assembly and cytoskeletal organisation[42]. As expected, depletion of either PIP4K2B or UHRF1 drove to strong impairment in focal adhesion number and actin organisation, which was followed by alterations in the capacity of the cells to exert forces on the substrates, spread and properly move. Again, this was not reversed by expression of super-active nuclear YAP mutant.

PIP4Ks are highly druggable lipid kinases recently proposed as possible targets for the treatment of different cancer types. During the last years, the rising interest in these enzymes led to the synthesis of different inhibitors targeting this family of phosphotransferases. Here, we tested two compounds, A131 and THZ-P1_2[45,46], to understand if direct inhibition of the kinase activity of PIP4K could recapitulate the phenotypes observed upon PIP4K2B silencing. Interestingly, both compounds altered UHRF1 expression, nuclear and cell mechanics, finally impacting on cell migration efficiency. This strongly suggests that PIP4K activity is strongly linked to the cell phenotypes described in this study.

In conclusion, our results prove that PIP4K2B is mechanosensitive. In fact, PIP4K2B depletion alters nuclear mechanics and impact on chromatin organisation, through a pathway involving UHRF1, finally altering YAP signalling and cell motility.

Since UHRF1 is highly expressed in cancer, while YAP signalling is known to trigger cancer initiation and growth of solid tumours, our study emphasises the possible benefits given by small compounds interventions targeting PIP4K as treatment for specific YAP-addicted cancer types.

## Methods
### Cell culture
hTERT_RPE1 cells (ATCC) were cultured in DMEM/F12 (Sigma Aldrich) medium supplemented with FBS 10%, Penicillin/Streptomycin (1X) and Glutamine (1X). HEK293T (ATCC), MEF (ATCC, and Hela (ATCC) cells were cultured in DMEM medium (Sigma Aldrich) supplemented with FBS 10%, Penicillin/Streptomycin and Glutamine. PC3 (ATCC) cells were cultured in RPMI1640 medium (Sigma Aldrich) supplemented with FBS 10%, Penicillin/Streptomycin and Glutamine. MDA-MB-231 (ATCC) were grown in Leibovitz's L15 (Sigma Aldrich) supplemented with FBS 10%, Penicillin/Streptomycin and Glutamine Inhibition of PIP4K was obtained treating cells with THZ-P1_2 (5/10 μM, MedChem express) or PIP4K-in-A131 (1/5 μM, MedChem. express) for 24 h. PI5P (20 μM) treatment was performed in cells growing in complete medium for 4 h, after a period of 16 h of serum starvation. MG-132 treatment (5 μM) was performed over-night in growing cells. Cell seeding on Fibronectin (20 μg/ml) or PDL (0.1 mg/ml) glass coated coverslips was performed as follows: glass coverslips were put in six-well plates filled with 1 ml of Fibronectin or PDL, and incubated for 1 h at room temperature. Coverslips were then washed three times with sterile PBS 1× and wells filled with complete medium. Cells were counted and seeded in order to reach 60-70% of confluency after 24 h for immunofluorescence experiments and 50% for cell motility experiments. For cell seeding onto hydrogels, hydrogels were coated with Fibronectin as reported above and cells cultured as indicated in the manuscript.

### Hydrogels preparation
Coverslips are prepared by rinsing in 70% ethanol followed by Plasma Cleaning. Coverslips are then activated to enhance hydrogel bonding. This is achieved through incubation of the coverslip with a 0.1 M NaOH solution. NaOH is allowed to evaporate and the NaOH-coated coverslip is then silanised with 0.5% (V/V) APTES (3-aminopropyl)trietholxysilane (SIGMA) in ddH2O for 5 min at room temperature. Coverslips are then washed three times in ddH2O to remove all remaining APTES and then allowed to air dry. Hydrogels were prepared in PBS using filtered Acrylamide and Bis-Acrylamide along with polymerizing agents Ammonium persulfate and TEMED, at 5% (V/V) each. For complete recipes of gels see Supplementary Table 1 in the Supplementary Information file. Gel solutions are quickly filtered and pipetted onto parafilm. APTES-coated coverslips were then inverted onto the gel solution and gel was allowed to polymerise. Following polymerisation coverslips were washed in 0.1 M HEPES pH 8.5 three times. In order to functionalise the gels, the polyacrylamide was activated with a 0.5 mM solution of sulfo-SANPAH(Thermo-fisher) prepared in 50 mM HEPES pH 8.5. Gels were covered in sulfo-SANPAH solution and exposed for 10 min to UV light (365 nm). The gel was washed once in 0.5 M HEPES pH 8.5 and the treatment was repeated again. Following the second treatment gels were washed three times to remove excess sulfo-SANPAH and then functionalised with fibronectin at 20 μg/ml. Coated coverslips were then washed in PBS and used for cell assays.

### Nuclear Envelope tension analysis and synthesis of cpst-FRET sensor
Nuclear Envelope (NE) tension was analysed using cpst[26] based FRET sensor built using Nesprin 1 protein as backbone, and named Mini-Nesprin 1. N-terminus Nesprin 1 (1521 bp, aa 1-507) and C-terminus (1422 bp, aa 8325−8797) coding sequences were synthesised exploiting IDT gBlocks Gene Fragments Technology. Briefly, gBlock 1 encoding N-terminus Calponin domains (CH-CH) of giant Nesprin 1 was cloned

into Spectrin-cpstFRET (plasmid#61109, Addgene) cut with AgeI and ScaI restriction enzymes (New England Biosciences). Then, cpst FRET probe from Addgene plasmid #61109 was PCR amplified and cloned between ScaI and KpnI restriction sites. Subsequently, gBlock 2 encoding C-terminus KASH domain was cloned between KpnI and MfeI restriction sites. A truncated version of the FRET sensor (CH-mutant) used as control in the setting-up of the experiments was obtained removing the first 751 bp from the original plasmid in order to brake CH domain and, then, impairing Actin binding of Mini_Nesprin 1. Graphical presentation of Mini_Nesprin 1 versions is reported in Supplementary Fig. 3F, G. Image acquisition was performed using a Leica TCS SP8 confocal microscope with Argon light laser as excitation source tuned at 458 nm and HC PL APO CS2 ×40/1.40 oil-immersion objective. Emission signals were captured every 10 nm starting from 460 nm up to 600 nm, and inverted FRET index calculated as the ratio between cpCerulean (~480 nm) and cpVenus (~530 nm) peaks. Data were analysed using in-house ImageJ macro.

## Chromatin compaction assay and FLIM analysis

Chromatin compaction analysis was performed as originally reported[30]. Briefly, cells were transfected using Neon electroporation system (Thermo Fischer) with pBabe_H2B-mCherry and pCDNA_H2B_GFP plasmids. Next, transfected cells were seeded on glass coverslips coated with Fibronectin (+/−hydrogels) and analysed using a Leica TCS SP8 confocal microscope with White light laser as excitation source tuned at 488 nm and HC PL APO CS2 ×40/1.40 oil-immersion objective, everything managed by Leica Application Suite X software, ver. 3.5.2.18963. For the lifetime measurements, the above system was implemented with PicoQuant Pico Harp 300 TCSPC module and picosecond event timer, managed by PicoQuant software (SymPho Time 64, ver. 2.4). Data were imported and analysed using in-house ImageJ macro.

## Traction force microscopy

Red fluorescent quantum dots (QDs) were deposited on silicone samples into confocal monocrystalline triangular arrays as described here[55]. In the experiment of TFM, hTERT_RPE1 cells were transduced and 4 days later seeded for 24 h on fibronectin-coated silicone (elastic modulus of 5.16 kPa) samples containing QDs. Cells were then analysed live using oil immersion 60x objective (Leica Germany) with Nikon Eclipse Ti microscope (Nikon Instruments). Reconstruction of the traction field from a single image of the displaced QDs exploited Cellogram software[55]. Displacement field to the QDs' resting position was inferred, and tractions calculated using the known material properties.

## Micro-Rheology analysis

Particle-tracking microrheology experiments were performed as previously described with minor changes. Carboxyl-modified fluorescent polystyrene particles (0.50 μm diameter, Polyscience) were introduced hTERT_RPE1 cells by using a ballistic gun (Bio-Rad). Helium gas at 900 psi was used to force a macro-carrier disk coated with particles to crash into a stopping screen. The force of collision was transferred to the particles, causing their dissociation from the macro-carrier and the bombardment of cells. Once bombarded, cells were extensively washed with PBS and left to recover for 24 h in culture medium. After recovery, cells were seeded ($10^4$ cells·cm$^{-2}$) on 35 mm glass bottom Petri dishes (Ibidi). After 24 h from cell seeding, the motion of nanoparticles embedded inside the cells was recorded for a total of 5 s at 100 frames per second using an inverted fluorescence microscope (Olympus IX81; Olympus) equipped with a 100× oil immersion objective (N.A. = 1.40), plus 1.6× magnification of internal microscope lens, and a Hamamatsu ORCA-Flash 2.8 CMOS camera (Hamamatsu). Experiments were performed under physiological conditions, using a microscope stage incubator (Okolab) to keep cells at 37 °C with 5% $CO_2$. The total number of analysed particles was at least 80 from more

than 10 cells for both cell lines. Particle-tracking microrheology allows to have indirect information about the local viscoelastic properties of living cells with a high spatiotemporal resolution, collecting and analysing the Brownian motions of particles embedded in the cytoplasm. By using our self-developed Matlab cose, the point tracking trajectories were generated in two distinct steps: firstly, the beads are detected in each frame, and then the points are linked into trajectories. Each position was determined by intensity measurements through its centroid, and it was compared frame by frame to identify the trajectory for each particle, based on the principle that the closest positions in successive frames belong to the same particle (proximity principle). Once the nanoparticle trajectories had been obtained, mean squared displacements (MSDs) were calculated from the following equation:

$$\langle \Delta r^2(t) \rangle = \langle [x(t - \tau)]^2 + [y(t - \tau)]^2 \rangle$$

where angular brackets mean time average, $\tau$ is the time scale and t is the elapsed time.

## Cell transfection, lentiviral production and cell transduction

Cell transfection was performed using Neon Transfection System (Thermo Fischer). hTERT_RPE1 cells were transfected using the following parameters: Voltage: 1350 V, Width: 20 ms, Number of pulses: $2 \times 10^6$ cells were transfected with ING2-PHD_GFP (Addgene, #21589) or pCSDEST2_NLS-GFP plasmids. Lentiviral particles were produced as described here[38]. Briefly, $3 \times 10^6$ HEK293T cells were transfected with pLKO_1 (encoding sh_RNAi) or pCMV6-AC (Origene #RG217766, encoding GFP-UHRF1) lentiviral plasmids, together with psPax2 and pMDG2 vectors with a ratio of 4 μg:2 μg:1 μg. Polyethylenimine (PEI) was used as transfection reagent. 24 h later medium was removed, and virus collection started after 24/48/72 h. Viral aliquots were pulled together and filtered with 0.45 μm filters. $2 \times 10^5$ cells were transduced with 1 ml of fresh virus through 20' of spinoculation at 2000rpm, then seeded in 6 well plates. sh_RNAi sequences are reported in the Supplementary Information file in Supplementary Table 2.

## Western blotting

Cell lysates were quantified analysing absorbance of the samples (595 nm) diluted in Bradford (Biorad) at spectrophotometer. They were next separated on bis-Tris sodium dodecyl sulfate–polyacrylamide gel electrophoresis gels, electroblotted to nitrocellulose, blocked in PBS–Tween-20 (0.1%) dried milk (5%), and then incubated with the primary antibodies. After washing, blots were incubated with appropriate secondary antibodies and developed using ECL-Plus (Biorad) at Chemidoc (Biorad). Quantification was performed through Fiji as the ratio between PIP4K2B and Tubulin (housekeeping) signals. A complete list of antibodies exploited in this work is available in Supplementary Table 3 in the Supplementary Information file, and full scan of gels is available in the Source Data file.

## Ubiquitination assay

Cells overexpressing UHRF1 were treated over-night with MG-132 at 5 μM as final concentration. Next, cells were lysed with RIPA buffer supplemented with MG-132 1 μM, DUB inhibitor PR-619 and beta-Mercaptoethanol. 500 μg of cell lysates immunoprecipitated using UHRF1 antibody (Santa Cruz) following manufacturer's protocol. Protein A/G coated beads were added to the mix, and everything was incubated over-night a 4 C in rotation. The next day, beads were washed five times with lysis buffer and resuspended with 50 μl Loading buffer 4X (Thermo Fischer) plus Reducing Agent 1X (Thermo Fischer). Western blotting was then performed, and samples were immunoblot with Ubiquitin antibody. The amount of UHRF1 immunoprecipitated was analysed using 5 μl of immunoprecipitated lysates, while inputs were loaded and immunoblotted a part in order to control quality of equal loading of different samples (using Tubulin as housekeeping).

## RT-qPCR

RNA was extracted with RNA extraction kit (Invitrogen), and quantified at Nanodrop. 100 ng of RNA were retrotranscribed into cDNA using a Reverse-Transcription kit (Life Technologies). qRT-PCR was performed using SYBR Green (Life Technologies) to detect expression of PIP4K2B, and TaqMan Master Mix (Invitrogen) for YAP target genes. GAPDH was used as housekeeping control (primers sequences and probes ID reported in Supplementary Table 5).

## RNA sequencing

For each experimental sample, totRNA was extracted using the RNeasy Prep kit (QIAGEN), and its abundance was measured using Qubit 4.0 and integrity assessed using Agilent Bioanalyzer 2100 using Nano RNA kit (RIN > 8). For each sample, an indexed-fragment library was prepared starting from 500 ng totalRNA using Illumina Stranded mRNA Prep ligation kit (Illumina) according to the manufacturer's instructions. Indexed libraries were quantified with Qubit HS DNA kit and controlled for proper size using Agilent Bioanalyzer 2100 High Sensitivity Dna kit prior to normalisation and equimolar pooling to perform a multiplexed sequencing run. 5% of Illumina pre-synthesised PhiX library was included in the sequencing mix, to serve as a positive control. Sequencing was performed in Paired End mode ($2 \times 75$ nt) on Illumina NextSeq550Dx instrument, generating on average 60 million PE reads per each library. Reads were aligned to the GRCh38/hg38 assembly human reference genome using the STAR aligner (v 2.6.1d).

Differential gene expression analysis was performed using the Bioconductor package DESeq2 (v 1.30.0) that estimates variance-mean dependence in count data from high-throughput sequencing data and tests for differential expression exploiting a negative binomial distribution-based model. Pre-ranked gene set enrichment analysis (GSEA) for evaluating pathway enrichment in transcriptional data was carried out using the Bioconductor package fgsea (v 1.16.0) and GSEA software taking advantage of the Reactome, KEGG, oncogenic signature and ontology gene sets available from the GSEA Molecular Signatures Database (https://www.gsea-msigdb.org/gsea/msigdb/genesets.jsp?collections). The RNA-seq dataset generated during the current study is available in the GEO (Gene Expression Omnibus) repository, under accession GSE200206.

## Cell motility assay

$3 \times 10^5$ hTERT_RPE1 cells were seeded in six-well plates over-night before live-microscopy analysis. Then, cells were stained with NucBlue Live Cell Reagent (Thermo Fischer) and imaged every 10 min for 18 h using a humidity- and temperature-controlled inverted wide-field microscope within an environmental chamber (ScanR, Olympus). Nuclear tracking and cell motility analysis were performed as described here[20]. For this purpose, specific software in C++ with the OpenCV [http://opencv.willowgarage.com/wiki/] and the GSL [http://www.gnu.org/software/gsl/] libraries were developed. The migration analysis was performed by the C++ software coupled with R [www.R-project.org].

## Cell spreading

Cells at different experimental conditions were trypsinised and resuspended in 1x Ringer buffer (150 mM NaCl, 1 mM MgCl2, 1 mM CaCl2, 20 mM Hepes (pH 7.4), 5 mM KCl and 2 g/l glucose). Suspended cells were seeded on fibronectin-coated glass coverslips (20 µg/ml) directly on the microscope stage at temperature-controlled conditions. Imaging was performed in DIC mode on the Leica AM TIRF MC microscope by a HCX PL APO 63×/1.47NA oil immersion objective. Cell spreading was followed for 30 min up to 1 h at a frame rate of 2'. Analysis of spreading capacity was calculated as spreading ration

between time 0 and time 30' for hTERT_RPE1 cells, and from 0' to 6 h for PC3 and MDA-MB-231 cells.

## Transmission electronic microscopy (TEM)

Electron microscopic examination was performed as previously reported[20]. A detailed description is explained below. EM based Morphological analysis: RPE1 cells grown on MatTek glass-bottom dishes (MatTek Corporation, Cat. P35G-1.5-14-C) for 24 h. Samples were fixed with of 4% paraformaldehyde and 2,5% glutaraldehyde (EMS) mixture in 0.2 M sodium cacodylate pH 7.2 for 2 hat RT, followed by 6 washes in 0.2 sodium cacodylate pH 7.2 at RT. Then cells were incubated in 1:1 mixture of 2% osmium tetraoxide and 3% potassium ferrocyanide for 1 h at RT followed by 6 times rinsing in cacodylate buffer. Then the samples were sequentially treated with 0.3% Thiocarbohydrazide in 0.2 M cacodylate buffer for 10 min and 1% OsO4 in 0.2 M cacodylate buffer (pH 6,9) for 30 min. Then, samples were rinsed with 0.1 M sodium cacodylate (pH 6.9) buffer until all traces of the yellow osmium fixative have been removed, washed in de-ionised water, treated with 1% uranyl acetate in water for 1 h and washed in water again. The samples were subsequently embedded in Epoxy resin at RT and polymerised for at least 72 h in a 60 °C oven. Embedded samples were then sectioned with diamond knife (Diatome) using Leica ultra-microtome. Sections were analysed with a Tecnai 20 High Voltage EM (FEI) operating at 200 kV.

## Immunofluorescence

Cells were seeded on Fibronectin (20 µg/µl) or PDL (0.1 mg/ml) coated coverslips (+/− hydrogels) for the time indicated in the experiments. Hydrogel preparation was performed using recipes reported in Supplementary Table 1 in the Supplementary Information file. Next, they were washed in PBS 1×, fixed with 4% paraformaldehyde for 10' and washed again three times with PBS 1×. Fixed cells were permeabilised with Triton 0.2% for 10', and incubated for 1 h in BSA 0.2%. Primary antibodies staining was performed in a humidity-controlled dark chamber at room temperature for 2 h and removed by three times of PBS washing. Secondary antibodies were used in dark at room temperature for 1 h, and nuclei stained with 4′,6-diamidino-2-phenylindole (DAPI, Sigma-Aldrich Cat. D8417) for 5 min. Samples were prepared using Glycerol as mounting medium and analysed. A complete list of the antibodies employed in this study is available in Supplementary Table 4 in the Supplementary Information file. Analysis of FA number, Actin organisation, and YAP nuclear/cytoplasmic ratio were performed exploiting custom Fiji scripts. For experiments intended to analyse nuclear size and shape, dimensionless parameter excess of perimeter (EOP) was calculated as the ratio of internal and perimetral length of the NE (as defined by Lamin A/C staining) over the perimeter of the convex envelope.

## Nuclear polarity analysis

Nuclear polarity analysis was performed as described here[20]. Briefly, cells were seeded on fibronectin-coated micro-patterned lines 10 µm. Patterns were fabricated using photolithography. The glass surface of the coverslips was treated and activated with plasma cleaner (Harrick Plasma), then coated with PLL-g-PEG (Surface Solutions GmbH, 0.1 mg/mL in 10 mM HEPES). After washing in PBS 1×, surface was illuminated with deep UV light (UVO Cleaner, Jelight) through a chromium photomask (JD-Photodata). Coverslips were then incubated with Fibronectin (20 µg/ml), and cells seeded over-night. Cell polarisation was assessed using Golgi staining marker Giantin (front), and IF was performed for the analysed NE components. Maps of nuclear polarity were created using a custom Fiji macro which quantifies the top 20% of the signal of the antibodies exploited in the IF. Images were taken every 0.6 µm of focal plane using z-stack function using oil immersion ×40/×60 objective (Leica Germany)

at Nikon Eclipse Ti microscope (Nikon Instruments) equipped with the UltraVIEW VoX spinning-disc confocal unit (PerkinElmer) and Velocity software (PerkinElmer).

## Cell cycle analysis

Cells growing in complete medium where collected, washed in PBS, and fixed in cold EtOH 70% overnight. Cells were washed with PBS and stained with propidium iodide (Abcam) and analysed by FACS.

## Cell apoptosis analysis

Cells were resuspended in Annexin V resuspension buffer and stained with Annexin V-FITC (BioLegend) for 20 min at room temperature and then directly analyzed by FACS.

## Cell proliferation

Cells were seeded at the same number in 24 wells ($1 \times 10^4$) in complete medium and counted manually for 3 days (24 h/48 h/72 h after seeding).

## Statistical analysis

Statistical analysis war performed using Unpaired *t*-test with Welch's Correction, and two-sided Kolmogorov–Smirnov test for nuclear polarity maps distribution. Every experiment was performed at least two times, independently. Statistical significative data are considered data with a pValue < 0.05.

## Reporting summary

Further information on research design is available in the Nature Portfolio Reporting Summary linked to this article.

## Data availability

The RNA-seq dataset generated during the current study is available in the GEO (Gene Expression Omnibus) repository, under accession GSE200206. Other data generated or analysed during this study are included in the Supplementary Information and are available from the corresponding authors upon request. Source data are provided with this paper.

## Code availability

Custom-built ImageJ macros and R scripts are available from the corresponding author upon request.

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

## Acknowledgements

This project was supported by Italian Association for Cancer Research (AIRC) Investigator Grant (Paolo Maiuri, #24976) and individual fellowship (Fabrizio A. Pennacchio, #23966). Alessandro Poli's work was founded by Fondazione Umberto Veronesi Post-doctoral fellowships (#000359) and Short-EMBO Fellowship (#8386). Polish National Science Centre grant founded Paulina Nastaly (#2020/39/D/NZ3/00882). We thank Kristina Havas, Giorgio Scita and Simona Polo for suggestions and discussions. RNA-sequencing and RT-qPCR analyses were performed by Cogentech. The authors would like to also thank IFOM Imaging Facility for help with performing experiments, Dr. Carlos Nino for help in setting the Ubiquitination assay, Giorgio Scita's group for providing pCDNA3.1_H2B-GFP, pBabe_H2B-mCherry, pLX304_YAP1-(S6A) (https://doi.org/10.1073/pnas.1703096114) and pCSDEST2_NLS-GFP, Kristina Havas' team for providing hydrogel preparation protocols and PI5P, and Maria Cristina Patroni Griffi for concept visualisation of Fig. 1F (mariacristina.patronigriffi@gmail.com).

## Author contributions

A.P. and P.M. designed experiments, supervised the project, interpreted and analysed the data. A.P., F.A.P., M.d.G. and M.L. performed biochemical experiments. A.P., A.G., M.L., P.N. and B.S. performed fixed and live cell imaging. P.N. produced probability density maps. F.A.P., S.F. and V.P. performed, interpreted and analysed cell rheology assay under the supervision of P.N. A.P. and F.P. performed, interpreted and analysed traction force microscopy data under the supervision of A.F and D.P. A.P., P.M. and M.C. analysed imaging data and carried out statistical analysis. A.P. generated the Mini Nesprin1 FRET sensor. A.P. and F.I. analysed RNAseq data. G.B. and A.M. performed electron microscopy experiments. A.F., N.G., P.N. and N.D. helped in data interpretation. A.P. and P.M. wrote the paper.

## Competing interests

The authors declare no competing interests.
