## [Peer Review File · Nature Communications]

PIP4K2B is mechanoresponsive and controls heterochromatin-driven nuclear softening through UHRF1REVIEWER COMMENTS

Reviewer #1 (Remarks to the Author):

The manuscript by Poli and collaborators "PIP4K2B is a mechanosensor and induces heterochromatin-driven nuclear softening through UHRF1" identifies the lipid kinase PIP4K2B as a novel mechanosensor involved in nuclear polarity and mechanics, as well as chromatin organization, through UHRF1. They finally aim at demonstrating that PIP4K2B activation/depletion has an impact on YAP signalling.

PIP4K2B is known to be involved in DNA integrity, by contributing to the expression of genes which are important to contro DNA damage, chromatin remodeling and protein-histone binding.

The study is of potential interest, since PIP4K2B is a druggable molecule whose expression has been associated with negative tumor prognosis. In this reviewer's opinion, this is the most noteworthy aspect of the manuscript.

The experimental design is clear and the experiments are well executed. Generally speaking, the conclusions are supported by the results of the experiments.

Few typos were spotted throughout the text, which should be fixed.

I have few concerns, which could be addressed through a limited amount of experiments and make the story more solid.

1. The authors demonstrate PIP4K2B levels are affected by substrate mechanics, but the mechanism responsible for the depletion of the protein remains unanswered. It would be interesting to know whether the Rho/ROCK pathway or F-actin integrity are involved in this process. In this case, cells treated with inhibitors of these axes should phenocopy the effects of soft substrate on PI4K2B protein expression.

2. The authors try to answer the question whether YAP exclusion from the nucleus happens before or after PIP4K2B depletion. Also, they question whether PIP4K2B activity (on gene expression, focal adhesion assembly, UHRF1 regulation, etc.) is mediated by YAP transcriptional activity. They decided to address this question by designing time-course experiments and - in fact - managed to show that YAP exclusion from the nucleus occurs after PIP4K2B depletion and follows UHRF1 regulation. These experiments are highly suggestive, but not conclusive. A simple experiment to answer whether PIP4K2B activities require YAP would significantly improve the impact of the story (maybe by expressing nuclear hyperactive mutant of YAP (YAP-S127A) in cells depleted for PIP4K2B).

3. Figure 1G shows mini-nesprin1 FRET representative images. The nuclear size appears strikingly different in silenced cells and compared to cells grown onto soft substrate, this impression being confirmed by FLIM images in Figure 2A. Reduced nuclear tension is, in fact, expected to cause changes in nuclear area. It would be interesting to see a graph where nuclear area is quantified in PIP4K2B-depleted cells.

5. The authors show that - regardless of the RNA-seq results - the protein levels of SUV39H1 were not altered in PIP4K2B-depleted cells. Can the authors speculate on the possible mechanism in the discussion?

Minor concerns:

Page 5, line 5: "rescueS" instead of "rescue".

Figure 5b-d: the labels should indicate the unit reported in the graphs. This applies to all the graphs in the manuscript.

Page 14, line 5: "...the effect seemed independent by pathways..." should be "...independent FROM...".

Page 14, line 21: the expression "As further confirm of this" seems to make no sense.

Page 15, line 15: Figure 4E instead of Figure 5E.

Page 19, line 15: "PIP4K2B lack drives (to) UHRF1 degradation

Page 19, line 21: "...independently OF...".

Reviewer #2 (Remarks to the Author):

In this study Poli et al identify PIP4K2B as a mechanically regulated kinase that changes protein expression levels in response to changes in ECM/substrate stiffness. The authors provide a lot of mechanistic insight into how this protein is regulated, and the downstream biochemical and mechanical signaling controlled by this pathway, which appears to be especially important with regards to YAP signaling, and in a broader view important to cancer. Overall this is a nice study identifying a new potential regulator of cell responses to substrate stiffness. The biggest concern is the large volume of results, some of which are difficult to understand. While I have pointed out issues with individual figures below, I would recommend the authors consider trying to restructure this paper so that it is more succinct.

Major concerns:

1. There is so much information in the paper it is a challenging read. This was especially the case with the first figure which covers 4 pages of results text, 3 subheadings in results, and 3 supplemental figures. Perhaps the authors can focus on condensing the results section down for the first figure? Or alternatively splitting this figure apart?

Is there an opportunity for a summary panel in the final figure that shows the overall pathway? Overall the writing could be improved to be more succinct in the Results section (less speculation/discussion, just focusing on results).

2. Nesprin-1 FRET sensor:

A) The nesprin-1 sensor is novel. It is exciting that a new sensor (beyond nesprin-2) has been developed. However, the rationale for this sensor is not clear. Presumably nesprin-1 and nesprin-2 behave the same connecting actin to the LINC complex. But maybe the authors have a reason to focus on nesprin-1 vs nesprin-2 (is it the more abundant isoform in these cells?). This part is not clear because the authors only measured nesprin-1 levels (Supplemental Figure 2C). The authors need to include nesprin-2 and SUN2 in their analysis of LINC complex changes with shRNA knockdown of PIP4K2B.

B) The use of "inverted" FRET index is somewhat unusual (normally FRET is reported as measured), and I am not clear on how this calculation or scaling is being performed.

C) The CH mutant is an important control that confirms the sensor is working. An additional control of actomyosin inhibition is typically included with new FRET sensors. Although the authors may be indirectly modulating actomyosin contractility on the softer 2.3 kPA gels, this is not as direct as treating cells with Y-compound and/or ML7. This is also important given the CPST choice of the linker is not as widely used as the flageliform TSmod and HP35 linkers which have been shown to be elastic and have validated FRET-force responsiveness (validated with optical tweezers).

D) I am also not clear if the other cells in Figure 1G are on FN-coated glass? If so, can the authors clarify this in the figure legend and also comment on the FRET value differences in this figure as compared to supplemental figure 3H, in which the cells have lower inverted FRET.

3. I wasn't understanding the plots in supplemental figure 2b and supplemental figure 2e. This figure legend says they reference Figure 2F, but there is no panel F in figure 2. Also the blue dotted line represents what? In the first column the word nucleus is blue, but I don't think that is what the blue line represents. Does the red line represent always the control? In some cases (lamin A/C, nesprin1) the red lines appear to be the same across all 3 panels, in other the red lines change. The authors need to clarify what these plots are showing and better explain the meaning of each color.

4. Figure 1F, can the front and rear positions of the nucleus be indicated on the figure? (e.g. is top

the front?)

5. Supplemental Figure 2D: Also needs clarification on front vs rear, as well as explanation on the 2 different heat maps given for the nucleus. I was not able to understand what is the difference between the 2 heatmaps of the nucleus in each condition. I also did not understand the similar maps in Supplemental Figure 3B.

6. Figure 2C: The authors state that the shRNA cells have less heterochromatin, but this was difficult to determine from the EM image (EM imaging is out of my expertise, however). Did the authors consider using histone acetylation/methylation markers to confirm this finding? Also the authors mention invaginations but show only 1 EM picture. Can these invaginations be seen with immunofluorescent staining? If so can the authors show how this looks across an entire nucleus?

7. Supplemental Figure 4B: The authors state this change in NLS localization of GFP to be mechanical regulation, but there are many other possibilities regarding this. The Roca-Cusachs group has suggested there are select conditions under which proteins are mechanosensitive to nuclear import, which depends on protein size. Since GFP is small this may not be the same situation as these publications. Additionally GFP alone (without NLS) is small enough to accumulate in the nucleus, so a small protein is not well suited to study active transport since it may also experience passive transport. Also there needs to be some discussion of the specific NLS used, since Roca-Cusachs has reported only some NLS sequences being mechanosensitive. While all of the data is very interesting (and I personally believe in the Roca-Cusachs work), it is my opinion that the authors conclusion that these data "confirm proper nuclear mechanics is important for active nuclear import of YAP" is overstated. I would encourage the authors to consider alternate possibilities as to how this protein could be affecting nuclear transport.

Reviewer #3 (Remarks to the Author):

In this study, the authors demonstrate that Phosphatidylinositol-5-phosphate (PtdIns5P)-4-kinases PIP4K2B is a key component of nuclear mechanosensing to tensile stress that translate into a change in cell morphology and motility. PIP4K2B generates PI5P in particular at the nuclear membrane. The study is well-written, well performed, and deploys a wide range of innovative tools to validate their findings. PIP4K2B depletion alters nuclear mechanics and acts on chromatin organization, nuclear localisation system, through a pathway involving UHRF1, next altering YAP signalling and cell motility. The tools related to the study of phosphoinositide nuclear signalling are relevant and include the validation of the specificity of this pathway to an isoform of PIP4K. Those data are pertinent to the field. However, prior publication, below are listed few points that are in my opinion needed to validate those findings as well assess their breadth in cell biology.

Major points :

- My main point is about translation of this work in other cell lines and cancer cell lines that overexpress PIP4K2B. The added data should include a validation on a functional assay such as motility assay on soft/rigid substrates.
- If the authors have indication that PIP4K2 nuclear overexpression is also occurring in tumour cells at the edge of the tumour limit that sense stiff environment, this data would add to the breadth of the data.
- In my opinion, while the data shown are convincingly demonstrating that PIP4K2 is mechanosensitive and takes part in the mechano-transduction pathway, it is not clear that PIP4K2 convert mechanical forces into biochemical forces, hence the title and conclusions throughout the whole manuscript should be revised accordingly.
- Confirmation of YAP regulation by analysis in time of several YAP-controlled gene expression, as well as validation of several targets from the RNAseq experiment is missing.
- Data on figure S7b are overinterpreted as only one time point is shown, it is unclear whether this experiment was repeated several times; also it seems to me that pAkt and pS6 are inversely regulated indicative of a feedback regulatory loop.

- The demonstration of specificity and efficiency of the inhibitors used in Fig6 is missing.
- Some statistical analysis are unclear to me (e.g. whether the authors are comparing individual cells from the same experiment or in independent experiments); also the numbers of independent replicates for each panel including in supplementary figures are not indicated for every panel.
- Controls on DAPI expression in all nuclear expression including Fig4B.
- Figure 4C : what happens to YAP localisation under PI5P treatment at later time points.

Minor points :

- Provide a clearer description in the introduction of the current knowledge about how mechanotransduction happens directly in the nucleus when cells are grown on soft / rigid substrate as it is not clear to the general reader. It is all the more important as the data shown are suggesting that mechanotransduction at the nuclear side precedes the membrane mechanotransduction (see also my comment above, as to me this conclusion could be strengthened).
- Describe in more details the coating of slides, plates and the hydrogel used as it is crucial to repeat the experiments.
- Spelling mistake : « Assesed »
- I do not understand this sentence « the effect seemed independent by pathways often involved in PIP4K signalling, »

Reviewer #1 (Remarks to the Author):

The manuscript by Poli and collaborators “PIP4K2B is a mechanosensor and induces heterochromatin-driven nuclear softening through UHRF1” identifies the lipid kinase PIP4K2B as a novel mechanosensor involved in nuclear polarity and mechanics, as well as chromatin organization, through UHRF1. They finally aim at demonstrating that PIP4K2B activation/depletion has an impact on YAP signalling. PIP4K2B is known to be involved in DNA integrity, by contributing to the expression of genes which are important to control DNA damage, chromatin remodeling and protein-histone binding. The study is of potential interest, since PIP4K2B is a druggable molecule whose expression has been associated with negative tumor prognosis. In this reviewer's opinion, this is the most noteworthy aspect of the manuscript. The experimental design is clear and the experiments are well executed. Generally speaking, the conclusions are supported by the results of the experiments. Few typos were spotted throughout the text, which should be fixed.

We thank the reviewer for the overall positive judgment of our work and for the interesting suggestions.

I have few concerns, which could be addressed through a limited amount of experiments and make the story more solid.

1. The authors demonstrate PIP4K2B levels are affected by substrate mechanics, but the mechanism responsible for the depletion of the protein remains unanswered. It would be interesting to know whether the Rho/ROCK pathway or F-actin integrity are involved in this process. In this case, cells treated with inhibitors of these axes should phenocopy the effects of soft substrate on PIP4K2B protein expression.

We really appreciated this suggestion and we tested the levels of PIP4K2B in hTERT-RPE1 cells upon treatment with three different compounds: Y-27632 (ROCK inhibitor), Blebbistatin (Myosin II inhibitor) and Latrunculin (inhibition of actin polymerization). Cells were treated with these drugs for 3h, then lysed and WB performed to assess PIP4K2B levels. Interestingly, as shown in the panel below, we found a strong decrease in PIP4K2B levels only when cells are treated with Latrunculin. These unexpected results suggest that cell contractility, inhibited with Y-27632 or Blebbistatin, is not involved in the control of PIP4K2B protein levels. On the contrary, PIP4K2B levels were affected by actin perturbations (Latrunculin treatment), suggesting a strong and direct connection between actin polymerization and these PI kinases. These findings are in line with previous reports on PI5P, the PIP4K substrate, which was described as a positive regulator of actin cytoskeletal rearrangements (Devereaux & Di Paolo, 2013 - doi: 10.1038/embor.2013.2). We could then hypothesize that cells respond to an excess of actin monomers (as induced by Latrunculin treatment or soft substrate) down-regulating PIP4K in order to

increase PI5P levels as feedback loop to push actin re-polymerisation and cytoskeletal rearrangement. These results are particularly interesting and pave the way to the understanding of the mechanism behind PIP4K2B degradation in cells growing on soft substrates. However, since they are not conclusive and much more would be needed to experimentally prove this hypothesis, we decided to show these new data and the relative discussion only in the response to reviewers (if not strongly suggested differently).

2. The authors try to answer the question whether YAP exclusion from the nucleus happens before or after PIP4K2B depletion. Also, they question whether PIP4K2B activity (on gene expression, focal adhesion assembly, UHRF1 regulation, etc.) is mediated by YAP transcriptional activity. They decided to address this question by designing time-course experiments and - in fact - managed to show that YAP exclusion from the nucleus occurs after PIP4K2B depletion and follows UHRF1 regulation. These experiments are highly suggestive, but not conclusive. A simple experiment to answer whether PIP4K2B activities require YAP would significantly improve the impact of the story (maybe by expressing nuclear hyperactive mutant of YAP (YAP-S127A) in cells depleted for PIP4K2B).

We thank the reviewer for this suggestion, it is a great idea. To implement it we took advantage of a hyperactive mutant of YAP, YAP-S6A, that is constitutively nuclear, and we expressed it in hTERT-RPE1 cells depleted for PIP4K2B. While in control cells, as expected, YAP-S6A induces an increase of UHRF1 protein level, in PIP4K2B depleted ones, its expression is not sufficient to rescue the UHRF1 level (Figure 4B). Similarly, YAP-S6A expression in PIP4K2B depleted hTERT-RPE1 cells does not rescue either focal adhesion (FA) number or cell spreading dynamics (Supp Figure 7G and Figure 5F). All together these new data are perfectly in line with our previous findings suggesting a post-transcriptional effect of PIP4K2B

signalling on UHRF1. They also show that YAP transcriptional activity is not necessary or sufficient to rescue the strong alterations driven by lack of either PIP4K2B or UHRF1.

3. Figure 1G shows mini-nesprin1 FRET representative images. The nuclear size appears strikingly different in silenced cells and compared to cells grown onto soft substrate, this impression being confirmed by FLIM images in Figure 2A. Reduced nuclear tension is, in fact, expected to cause changes in nuclear area. It would be interesting to see a graph where nuclear area is quantified in PIP4K2B-depleted cells.

We quantified the nuclear area of cells depleted for PIP4K2B and control cells. Interestingly, as spotted by the reviewer, we found that cells lacking PIP4K2B were characterized by an increase in nuclear area. This is in line with mini-Nesprin1 FRET and FLIM representative images. We added a relative new plot in SuppFig 3H.

5. The authors show that - regardless of the RNA-seq results - the protein levels of SUV39H1 were not altered in PIP4K2B-depleted cells. Can the authors speculate on the possible mechanism in the discussion?

SUV39H1 and SUV39H2 are the main histone methyltransferases involved in H3K9 trimethylation. SUV39H1 protein stability and expression is augmented in conditions of oxidative/metabolic stress response (by SirT1, Bosch-Presegué et al., 2011 - <https://doi.org/10.1016/j.molcel.2011.02.034>). PIP4K depletion is known to trigger reactive oxygen species (ROS) and oxidative stress in particular cell conditions (Fiume et al., 2015 - DOI: <https://doi.org/10.1016/j.bbalip.2015.02.014>). Moreover, it was recently reported that cells growing on soft extracellular matrix are characterized by increased levels of ROS and oxidative stress (Romani et al., 2022 - DOI: [10.1038/s41556-022-00843-w](https://doi.org/10.1038/s41556-022-00843-w)). Altogether, we could speculate that lack of PIP4K2B could trigger oxidative stress in hTERT-RPE1 cells, an event which could counteract loss of transcription of SUV39H1 gene reinforcing protein stability and accumulation in cells. We added to the discussion section this:

“Surprisingly, while SUV39H1 mRNA levels were low both in soft-seeded and in PIP4K2B knock-down cells, its protein levels were not altered. This discrepancy could be due to an increased SUV39H1 protein stability as a consequence of oxidative stress, which is usually induced by soft substrates or by the depletion of PIP4K (Fiume et al., 2015 - DOI: <https://doi.org/10.1016/j.bbalip.2015.02.014>).”

Minor concerns:

Page 5, line 5: “rescueS” instead of “rescue”.

Figure 5b-d: the labels should indicate the unit reported in the graphs. This applies to all the graphs in the manuscript.

Page 14, line 5: "...the effect seemed independent by pathways..." should be "...independent FROM...".

Page 14, line 21: the expression "As further confirm of this" seems to make no sense.

Page 15, line 15: Figure 4E instead of Figure 5E.

Page 19, line 15: "PIP4K2B lack drives (to) UHRF1 degradation

Page 19, line 21: "...independently OF...".

We apologize for these mistakes and we corrected it in the revised version of our manuscript.

Reviewer #2 (Remarks to the Author):

In this study Poli et al identify PIP4K2B as a mechanically regulated kinase that changes protein expression levels in response to changes in ECM/substrate stiffness. The authors provide a lot of mechanistic insight into how this protein is regulated, and the downstream biochemical and mechanical signaling controlled by this pathway, which appears to be especially important with regards to YAP signaling, and in a broader view important to cancer. Overall this is a nice study identifying a new potential regulator of cell responses to substrate stiffness. The biggest concern is the large volume of results, some of which are difficult to understand. While I have pointed out issues with individual figures below, I would recommend the authors consider trying to restructure this paper so that it is more succinct.

We thank the reviewer for the positive comments. As suggested we shorten the manuscript and we tried to clarify it.

Major concerns:

1. There is so much information in the paper it is a challenging read. This was especially the case with the first figure which covers 4 pages of results text, 3 subheadings in results, and 3 supplemental figures. Perhaps the authors can focus on condensing the results section down for the first figure? Or alternatively splitting this figure apart?

Is there an opportunity for a summary panel in the final figure that shows the overall pathway?

Overall the writing could be improved to be more succinct in the Results section (less speculation/discussion, just focusing on results).

As suggested, we simplified Figure 1. We moved panel 1H in Figure 2 (now as 2.A). We also added a graphical explanation for cell polarization on linear patterns (see also later). We finally added a graphical summary which recapitulates the main results of the paper (Figure 7). Regarding the text we tried to simplify the results section and shorten the paper.

2. Nesprin-1 FRET sensor:

A) The nesprin-1 sensor is novel. It is exciting that a new sensor (beyond nesprin-2) has been developed. However, the rationale for this sensor is not clear. Presumably nesprin-1 and nesprin-2 behave the same connecting actin to the LINC complex. But maybe the authors have a reason to focus on nesprin-1 vs nesprin-2 (is it the more abundant isoform in these cells?). This part is not clear because the authors only measured nesprin-1 levels (Supplemental Figure 2C). The authors need to include nesprin-2 and SUN2 in their analysis of LINC complex changes with shRNA knockdown of PIP4K2B.

We thank the reviewer for the interest in our new FRET sensor. The rationale behind the selection of Nesprin1 over Nesprin2 was that this isoform is highly expressed in our cells of interest, while Nesprin2 is almost absent (as evident in the RNA seq data). Moreover, as requested, we immunoblotted for Nesprin2 and SUN2 hTERT-RPE1 cell lysates after depletion of PIP4K2B. Nesprin 2 was not detectable, while levels of SUN2 seemed unchanged upon PIP4K2B silencing (See supp. Figure 2C).

B) The use of "inverted" FRET index is somewhat unusual (normally FRET is reported as measured), and I am not clear on how this calculation or scaling is being performed.

The use of Inverted FRET index was meant to avoid confusion. Indeed, in this way the extent of tension on the NE is directly proportional to the Inverted FRET index, i.e. the higher is the value (then, in principle low FRET), the higher is the tension and vice versa (Meng & Sachs 2012 - <https://doi.org/10.1242/jcs.093104>). The Inverted FRET index was computed as the ratio between donor signal intensity (cpCerulean) and acceptor signal intensity (cpVenus).

C) The CH mutant is an important control that confirms the sensor is working. An additional control of actomyosin inhibition is typically included with new FRET sensors. Although the authors may be indirectly modulating actomyosin contractility on the softer 2.3 kPA gels, this is not as direct as treating cells with Y-compound and/or ML7. This is also important given the CPST choice of the linker is not as widely used as the flageliform TSmold and HP35 linkers which have been shown to be elastic and have validated FRET-force responsiveness (validated with optical tweezers).

We thank the reviewer for this comment and as suggested we added a new control. In Supplementary Figure 3H we now also report inverted FRET index values for Latrunculin treated cells, that are similar to the ones measured in cells growing on soft substrates or coverslips coated with PDL.

D) I am also not clear if the other cells in Figure 1G are on FN-coated glass? If so, can the authors clarify this in the figure legend and also comment on the FRET value differences in this figure as compared to supplemental figure 3H, in which the cells have lower inverted FRET.

We apologize for this lack of clarity. We amended this part in the figure legend to properly indicate how cells were seeded (see new Figure 2A). Control cells, sh_Ctrl and Glass_FN, were both seeded on Fibronectin coated glass-coverslips. The differences in inverted FRET values between the 2 figures is probably due to differences in the previous treatment of the cells. In fact, Glass_FN cells correspond

to wild type hTERT_RPE1 cells, not previously transduced with shRNA (sh_Ctrl). It is important to state that every experiment in which the FRET sensor was used has always a proper internal control, like sh_Ctrl for sh_PIP4K2B.

3. I wasn't understanding the plots in supplemental figure 2b and supplemental figure 2e. This figure legend says they reference Figure 2F, but there is no panel F in figure 2. Also the blue dotted line represents what? In the first column the word nucleus is blue, but I don't think that is what the blue line represents. Does the red line represent always the control? In some cases (lamin A/C, nesprin1) the red lines appear to be the same across all 3 panels, in other the red lines change. The authors need to clarify what these plots are showing and better explain the meaning of each color.

We apologize for this mistake and for the lack of clarity. Plots in Supplementary Figure 2.B refer indeed to maps in Figure 1.F and not 2.F. We corrected it in the revised version of our manuscript. Plots in Supplementary Figure 2.E correctly refer to Supplementary Figure 2.D.

These plots represent front-to-rear normalized density corresponding to the relative proteins of interest distribution maps. As described in Nastaly et al (doi: 10.1038/s41467-020-15910-9), distribution maps are obtained by overlapping registered images; normalized density plots are then the corresponding front to rear cumulative probability function. The dotted blue line in the density plot should always be the control. In the first column of Supp_Fig 2.B we are comparing the distribution of proteins with respect to the uniform condition (the nucleus itself, that then is the diagonal) to test if there is a statistically significant bias of the protein distribution front versus back. In the other two columns we test one condition versus another. So in the first plot sh_Ctrl is shown as red, versus blue nucleus (DAPI). In the subsequent graphs, sh_Ctrl is blue and sh_PIP4K2B (#1 and #2) are red. We corrected it in the revised version of our manuscript. Also, the first panel of Supp. Figure 2B (sh_Ctrl vs Nucleus for Emerin) was wrong, we apologize for this trivial mistake and we really thank the reviewer for spotting it.

4. Figure 1F, can the front and rear positions of the nucleus be indicated on the figure? (e.g. is top the front?)

See also point 1: we agree with the reviewer that this is necessary to help readers to understand the meaning of the figure. We added it in this new version.

5. Supplemental Figure 2D: Also needs clarification on front vs rear, as well as explanation on the 2 different heat maps given for the nucleus. I was not able to understand what is the difference between the 2 heatmaps of the nucleus in each condition. I also did not understand the similar maps in Supplemental Figure 3B.

We are sorry and we added the front-rear legend to the maps. For each protein distribution map there is a corresponding nucleus (DAPI) map. The first is generated by the protein of interest signal, the latter by the DAPI one. The distribution of the DAPI is used as reference of uniformity and to normalize the probability cumulative function. In Supp. Figure 3B nuclear distribution map of Mini-Nesprin1 is shown together with its own DAPI (nucleus) distribution.

6. Figure 2C: The authors state that the shRNA cells have less heterochromatin, but this was difficult to determine from the EM image (EM imaging is out of my expertise, however). Did the authors consider using histone acetylation/methylation markers to confirm this finding? Also the authors mention invaginations but show only 1 EM picture. Can these invaginations be seen with immunofluorescent staining? If so can the authors show how this looks across an entire nucleus?

Chromatin density was initially evaluated through EM, and then confirmed by reduction of heterochromatin marker H3K9me3 in cells depleted of PIP4K2B (sh_PIP4K2B #1 and sh_PIP4K2B#2) or growing in soft substrates (see Figure 2C). Moreover, transcriptomic data indicated silencing of PIP4K2B to negatively correlate with genes involved in heterochromatin (Figure 3C and Supp Fig5B, 5C and 5D) formation and maintenance. We now added new images of LMNA/C stained nuclei a relative quantification of nuclear invagination extent (Supp. Figure 4B and 4C).

7. Supplemental Figure 4B: The authors state this change in NLS localization of GFP to be mechanical regulation, but there are many other possibilities regarding this. The Roca-Cusachs group has suggested there are select conditions under which proteins are mechanosensitive to nuclear import, which depends on protein size. Since GFP is small this may not be the same situation as these publications. Additionally GFP alone (without NLS) is small enough to accumulate in the nucleus, so a small protein is not well suited to study active transport since it may also experience passive transport.

Also there needs to be some discussion of the specific NLS used, since Roca-Cusachs has reported only some NLS sequences being mechanosensitive.

While all of the data is very interesting (and I personally believe in the Roca-Cusachs work), it is my opinion that the authors conclusion that these data "confirm proper nuclear mechanics is important for active nuclear import of YAP" is overstated. I would encourage the authors to consider alternate possibilities as to how this protein could be affecting nuclear transport.

We partially agree with this reviewer comment. Indeed, as the reviewer pointed out, in the last work from Roca-Cusachs team (Mechanical force application to the nucleus regulates nucleocytoplasmic transport, NCB 2022) it is clearly shown that mechanosensitivity of nuclear import is a function of both protein size and NLS

sequence. Here, we measured a decreased NC ratio of GFP-nls upon PIP4K2B KD and we correlated it with the independent observation that in the same condition NE tension is decreased too. Nevertheless, it is not the purpose of our work to prove that cell tensional state affects nuclear/cytoplasmic transport, although our observations are evocative of it. For this reason, we decided to remove the data relative to GFP-nls from the manuscript and only refer to published data as one of the possible interpretations of our results. The paragraph now ends with: “Al together these observations show that lack of PIP4K2B impacts on fundamental nuclear mechanical properties, as NE tension and chromatin compaction, and it also impairs proper nuclear/cytoplasmic shuttling of YAP.”

Reviewer #3 (Remarks to the Author):

In this study, the authors demonstrate that Phosphatidylinositol-5-phosphate (PtdIns5P)-4-kinases PIP4K2B is a key component of nuclear mechanosensing to tensile stress that translate into a change in cell morphology and motility. PIP4K2B generates PI5P in particular at the nuclear membrane. The study is well-written, well performed, and deploys a wide range of innovative tools to validate their findings. PIP4K2B depletion alters nuclear mechanics and acts on chromatin organization, nuclear localisation system, through a pathway involving UHRF1, next altering YAP signalling and cell motility. The tools related to the study of phosphoinositide nuclear signalling are relevant and include the validation of the specificity of this pathway to an isoform of PIP4K. Those data are pertinent to the field. However, prior publication, below are listed few points that are in my opinion needed to validate those findings as well assess their breadth in cell biology.

We thank the reviewer for the positive comments regarding our study.

Major points :

- My main point is about translation of this work in other cell lines and cancer cell lines that overexpress PIP4K2B. The added data should include a validation on a functional assay such as motility assay on soft/rigid substrates.

We thank the reviewer for this comment and indeed we extended our study to other two cancer cell lines. In the first version of our paper we showed that in two normal cell lines, RPE1 and MEF, as well as in a cancer cell line, HeLa, PIP4K2B protein levels decrease when cells grow on soft substrates. Now we confirmed this observation in PC3 and MDA-MB-231 cell lines, models of prostate and breast cancer respectively. Interestingly, the dynamics of PIP4K2B drop in PC3 and MDA-MB-231 is slower than what was observed in RPE1, MEF or HeLa (Supp. Figure 1A). Moreover, we silenced PIP4K2B in both PC3 and MDA-MB-231 and analyzed: UHRF1 levels, spreading dynamics and YAP localization. We found that, as for RPE1, PIP4K2B depletion induces a drastic decrease in UHRF1 protein levels. Similarly, it also causes a significant reduction of cell spreading speed and a lack of focal adhesions. This new set of data are now part of a novel Supp. Figure 8A and B.

- If the authors have indication that PIP4K2 nuclear overexpression is also occurring in tumour cells at the edge of the tumour limit that sense stiff environment, this data would add to the breadth of the data.

We thank the reviewer for this interesting suggestion and we agree that this would significantly increase the impact of our work. It has already been reported that expression of PIP4K is strictly correlated with different types of cancer (Kaur Arora G. et al, 2021 - DOI:10.1002/1873-3468.14237; Fiume et al., 2015 - DOI:

<https://doi.org/10.1016/j.bbali.2015.02.014>), although precise localization in tissues is not perfectly understood.

We are sorry, but we have only limited access to tumor samples and we did not implement this suggestion. Since our manuscript is mainly focused on the mechanosensing property of PIP4K2B we decided to concentrate our efforts on improving other aspects of our work that we think are more in line with the scope of this paper.

- In my opinion, while the data shown are convincingly demonstrating that PIP4K2 is mechanosensitive and takes part in the mechanotransduction pathway, it is not clear that PIP4K2 convert mechanical forces into biochemical forces, hence the title and conclusions throughout the whole manuscript should be revised accordingly.

We only partially agree with the reviewer and indeed we do not sustain that PIP4K2B directly translates mechanical forces into biochemical signals (for instance we did not show changes in the levels of PI5P or other PIs).

However, we do refer to PIP4K2B as a novel “mechanosensor” since it is degraded as cellular response to soft substrate and since its direct depletion or inhibition is sufficient to well phenocopy this response. Moreover, we proved that PIP4K2B degradation effects are YAP independent. On the other hand, we showed that activity of PIP4K2B is strongly correlated with the phenotypes described (through use of PIP4K inhibitors and treating cells directly with PI5P).

To define as mechanosensors the only elements able to translate a mechanical stimulus into a biochemical signal, is really strict, although correct. However, in literature, proteins or transcription factors, whose concentration or localization changes in function of mechanical stimuli and that impact on mechanotransduction, are often generally called mechanosensors.

For all these reasons, we only slightly changed the title of our manuscript.

- Confirmation of YAP regulation by analysis in time of several YAP-controlled gene expression, as well as validation of several targets from the RNAseq experiment is missing.

We thank the reviewer for suggesting this control. We then added in the revised version of our paper RT-qPCR analysis of different YAP target genes, as validation of the previous RNAseq data (Supp. Figure 5A). They are all downregulated upon PIP4K2B silencing.

- Data on figure S7b are overinterpreted as only one time point is shown, it is unclear whether this experiment was repeated several times; also it seems to me that pAkt and pS6 are inversely regulated indicative of a feedback regulatory loop.

We are sorry for this possible overinterpretation and, moreover, we did not explore these pathways in a time-dependent manner. Since this data is marginal in the overall story and clearly incomplete, for the sake of clarity, we decided to remove it.

- The demonstration of specificity and efficiency of the inhibitors used in Fig6 is missing.

This is an important point and we clarified it in the new version of our manuscript. These compounds are described as pan PIP4K inhibitors, with THZ_P1 which preferentially inhibits PIP4K2A and 2B over PIP4K2C [Kitagawa M. et al, 2017 doi: 10.1038/s41467-017-02287-5. and Chen S. et al, 2021 doi:10.1073/pnas.2002486118]. As control of their efficiency we added WB analyses of UHRF1 in cells treated with the inhibitors. Treatment of hTERT_RPE1 cells with A-131 or THZ_P1 drives UHRF1 downregulation similarly to PIP4K2B depletion by ShRNAi.

- Some statistical analysis are unclear to me (e.g. whether the authors are comparing individual cells from the same experiment or in independent experiments); also the numbers of independent replicates for each panel including in supplementary figures are not indicated for every panel.

We are sorry for this lack of clarity. We amended this indicating in figure legends the details of data shown.

- Controls on DAPI expression in all nuclear expression including Fig4B.

We are sorry for the lack of this control, we added it in the new version of Figure 4B and Supp. Figure 6D.

- Figure 4C : what happens to YAP localisation under PI5P treatment at later time points.

This is an important and interesting point. Unfortunately, prolonged treatment of cells with PI5P led to cell death. For this reason, we decided to use it only for a short time period.

Minor points :

- Provide a clearer description in the introduction of the current knowledge about how mechanotransduction happens directly in the nucleus when cells are grown on soft / rigid substrate as it is not clear to the general reader. It is all the more important as the data shown are suggesting that mechanotransduction at the nuclear side precedes the membrane mechanotransduction (see also my comment above, as to me this conclusion could be strengthened).

We are sorry for this lack of clarity and we tried to better implement it in the introduction, highlighting recent reports on the role of the nucleus in response to mechanical stimuli. The tight connection between the nucleus and the cytoplasm allows instantaneous transmission of mechanically induced changes in cell shape or contractility status, to the nucleus, which then acts as a whole as a “mechanosensor”. Here, we identified a new player on the nuclear response to mechanical stimuli, PIP4K2B. We showed that its levels are a function of cell substrate stiffness and that its degradation is sufficient to induce chromatin state alterations then driving cell response.

- Describe in more details the coating of slides, plates and the hydrogel used as it is crucial to repeat the experiments.

We are sorry for this lack of clarity. We accordingly improved the Materials and Methods section and the Supp Materials.

- Spelling mistake : « Assesed »

Thank you for pointing it out.

- I do not understand this sentence « the effect seemed independent by pathways often involved in PIP4K signalling, »

We apologize, data shown in Supplementary Figure 7.B is incomplete and could lead to possible misinterpretation. Since it is not crucial for the main message of our paper and following criticism from another reviewer, we decided to remove it.

REVIEWERS' COMMENTS

Reviewer #1 (Remarks to the Author):

The authors addressed all my concerns.

Reviewer #2 (Remarks to the Author):

This revised manuscript by Poli et al has been greatly improved by the responsiveness of the authors to my critique as well as the other two critiques. My opinion is that the figures are much clearer and the findings easier to understand in the revised manuscript. As mentioned in my first review I found the findings to be significant and well supported.

I have one remaining minor critique, Figure 2C, the H3K9me staining is quite faint in the images (almost impossible to see in the PDF). I would recommend the authors adjust the image scaling (evenly across all images) to better showcase the differences.

Reviewer #3 (Remarks to the Author):

The authors have added some proofs that their findings are pertinent to other cell types and importantly to cancer cell lines (PC3 and MDA-MB-231 cell lines).

The authors have answered / discussed in a sufficient manner the questions raised; they have in particular added important details about the selectivity of PIP4K inhibitors.

In my opinion, this article is suitable for publication and provides a novel insight on the role of phosphoinositides in nuclear mechanotransduction.